# Hypoxia controls plasma membrane targeting of polarity proteins by dynamic turnover of PI4P and PI(4,5)P2

Juan Lu[†], Wei Dong[†], Gerald R Hammond*, Yang Hong*

Department of Cell Biology, University of Pittsburgh, Pittsburgh, United States

**Abstract** Phosphatidylinositol 4-phosphate (PI4P) and phosphatidylinositol 4,5-biphosphate (PIP2) are key phosphoinositides that determine the identity of the plasma membrane (PM) and regulate numerous key biological events there. To date, mechanisms regulating the homeostasis and dynamic turnover of PM PI4P and PIP2 in response to various physiological conditions and stresses remain to be fully elucidated. Here, we report that hypoxia in *Drosophila* induces acute and reversible depletion of PM PI4P and PIP2 that severely disrupts the electrostatic PM targeting of multiple polybasic polarity proteins. Genetically encoded ATP sensors confirmed that hypoxia induces acute and reversible reduction of cellular ATP levels which showed a strong real-time correlation with the levels of PM PI4P and PIP2 in cultured cells. By combining genetic manipulations with quantitative imaging assays we showed that PI4KIIIα, as well as Rbo/EFR3 and TTC7 that are essential for targeting PI4KIIIα to PM, are required for maintaining the homeostasis and dynamic turnover of PM PI4P and PIP2 under normoxia and hypoxia. Our results revealed that in cells challenged by energetic stresses triggered by hypoxia, ATP inhibition and possibly ischemia, dramatic turnover of PM PI4P and PIP2 could have profound impact on many cellular processes including electrostatic PM targeting of numerous polybasic proteins.

\*For correspondence:
ghammond@pitt.edu (GRH);
yhong@pitt.edu (YH)

[†]These authors contributed equally to this work

**Competing interest:** The authors declare that no competing interests exist.

## Editor's evaluation

The authors show that hypoxia leads to previously unappreciated effects on levels of plasma membrane PI4P and PIP2, which affects membrane targeting of proteins important for normal cellular physiology, including cell polarity. They provide insight into the identity of the PI4Ks that are responsible for regenerating plasma membrane PIP2 following return to normoxia. These findings are novel and provide multiple interesting insights for those studying phosphoinositide biology as well as cellular responses to hypoxic stress and recovery.

## Introduction

The inner leaflet of the plasma membrane (PM) is the most negatively charged membrane surface due to its enrichment of phospholipids including phosphatidylserine and phosphoinositides (PPIns) PI4P (phosphatidylinositol (PtdIns) 4-phosphate) and PIP2 (PtdIns 4,5-biphosphate (PI(4,5)P2)). The unique combination of PI4P and PIP2 is crucial to determine the PM identity by regulating many key biological events in the PM including cell signaling, endocytosis, and channel activation (*Hammond et al., 2012*). Moreover, for proteins with positively charged domains/surfaces, electrostatic binding to the PM is a fundamental mechanism underlying the regulation of their subcellular localization and biological activity (*McLaughlin and Murray, 2005*). One typical example can be found in polarity proteins that play essential and conserved roles in regulating various types of cell polarity such as apical-basal polarity in epithelial cells (*Bailey and Prehoda, 2015*; *Dong et al., 2020*; *Dong et al., 2015*; *Hong,*

*2018*; *Lu et al., 2021*). Recent discoveries from our group showed that multiple polarity proteins such as Lgl, aPKC, and Dlg contain positively charged polybasic motifs that electrostatically bind the negatively charged inner surface of PM (*Dong et al., 2020*; *Dong et al., 2015*; *Lu et al., 2021*), and such electrostatic PM targeting has now emerged as a mechanism essential for regulating their subcellular localization and biological activities in cell polarity.

While mechanisms regulating the interaction between polybasic motifs and PM have been relatively well studied (*Bailey and Prehoda, 2015*; *Dong et al., 2020*; *Dong et al., 2015*; *Hong, 2018*; *Lu et al., 2021*), much less is known how the homeostasis and turnover of PM PI4P and PIP2 may impact the electrostatic PM targeting. Although sophisticated mechanisms exist to maintain the steady state levels of PM PI4P and PIP2 under normal conditions (*Chen et al., 2017*; *Dickson et al., 2014*; *Wang et al., 2019*), our previous live imaging experiments in *Drosophila* showed a striking phenomenon that hypoxia induces acute and reversible loss of PM localization of polybasic polarity proteins Lgl, aPKC, and Dlg in epithelial cells (*Dong et al., 2020*; *Dong et al., 2015*; *Lu et al., 2021*), likely through reducing intracellular ATP levels (*Dong et al., 2015*). Our previous studies also showed that PM PIP2 could be reversibly depleted under hypoxia (*Dong et al., 2015*), suggesting that a potential connection from hypoxia to ATP inhibition to PM phospholipids depletion to loss of electrostatic PM targeting of polybasic proteins. However, to date how PM PI4P levels are regulated under hypoxia is unknown. Even less is known about the mechanisms through which hypoxia and ATP inhibition impact PM PI4P and PIP2 levels, and consequently the electrostatic PM targeting of numerous proteins.

In this report, we carried out quantitative live imaging experiments in *Drosophila* and cultured mammalian cells using multiple genetically encoded sensors to show that acute hypoxia induces dramatic but reversible depletion of PM PI4P and PIP2, accompanied by concurrent loss of PM localization of polybasic polarity protein Lgl. Using genetically encoded ATP sensors, we also confirmed a real-time correlation between the intracellular ATP levels and PM levels of PI4P and PIP2 in cultured cells. We further identified that PI4KIIIα (PtdIns-4 kinase IIIα) and its PM targeting machinery are required for the proper dynamic turnover of PM PI4P and PIP2 under hypoxia and ATP inhibition, as well as for properly restoring the post-hypoxia electrostatic PM targeting of Lgl. Our studies reveal a potential regulatory mechanism that dynamically controls PM PI4P and PIP2 levels in response to hypoxia and ATP inhibition. Our results suggest that genetic deficiencies in regulating such dynamic turnover of PM PI4P and PIP2 could have profound impact on cell physiology including polarity, when cells are challenged by energetic stresses triggered by hypoxia, ischemia and ATP inhibition.

## Results
### Hypoxia triggers acute and reversible loss of PM PI4P and PIP2

Based on a serendipitous observation that PM targeting of polybasic polarity protein Lgl appeared to be sensitive to hypoxia (*Dong et al., 2015*), we previously established custom live imaging assays (see below) to demonstrate that all three polybasic polarity proteins, Lgl, aPKC, and Dlg, showed acute and reversible loss of PM targeting under 30–60 min of hypoxia (0.5% $O_2$) in *Drosophila* follicle and embryonic epithelial cells in vivo (*Dong et al., 2020*; *Dong et al., 2015*; *Lu et al., 2021*). Since PM PIP2 also appeared to be transiently depleted under hypoxia in such assays (*Dong et al., 2015*), we decided to systematically investigate how hypoxia impacts the PM PI4P and PIP2 levels in vivo. We used follicular epithelial cells of *Drosophila* ovaries as the primary system as they are well established for ex vivo live imaging (*Prasad and Montell, 2007*) and for genetic manipulations such as RNAi knock-down.

We generated transgenic flies that ubiquitously express the PI4P sensor P4M × 2::GFP (*Sohn et al., 2018*) as well as PIP2 sensors PLCδ-PH::GFP and PLCδ-PH::RFP (hereafter referred as PLC-PH::GFP and PLC-PH::RFP, respectively) (*Wills et al., 2018*). Consistent with PI4P being mostly enriched at both PM and membranes of intracellular compartments such as endosomes and Golgi, P4M × 2::GFP can be seen in both PM and intracellular puncta in follicle epithelial cells (*Figure 1A*). To investigate hypoxia-induced turnover of PI4P and PIP2, ovaries dissected from flies expressing both P4M × 2::GFP and PLC-PH::RFP were mounted and imaged in custom micro chambers that can be flushed with either 0.5% $O_2$/99.5% $N_2$ gas mixture for hypoxia or normal air for reoxygenation. Within ~60 minutes of hypoxia, both PI4P and PIP2 sensors were gradually lost from PM, with PM PLC-PH::RFP diminished faster than PI4P (*Figure 1A*, *Figure 1—video 1*). Once the imaging chamber was reoxygenated

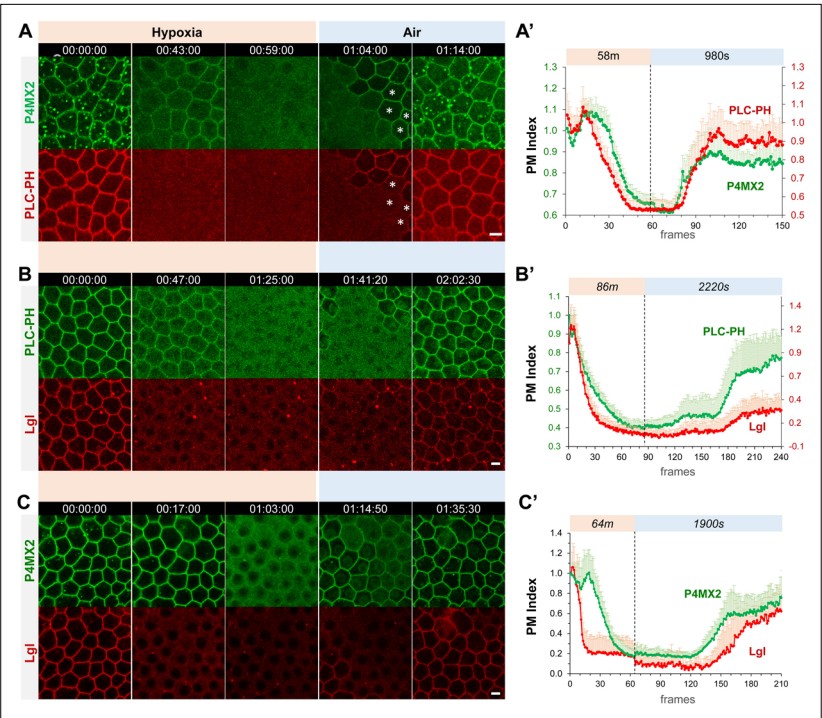

**Figure 1.** Hypoxia induces acute and reversible loss of P4M × 2::GFP and PLC-PH::RFP from the PM in *Drosophila* follicle cells. (**A–C**) Representative frames showing follicle cells coexpressing P4M × 2::GFP and PLC-PH::RFP (**A**) or P4M × 2::GFP and Lgl::RFP (**B**) or PLC-PH::GFP and Lgl::RFP (**C**) undergoing hypoxia and reoxygenation. In A, at 1:04:00, asterisks (*) highlight cells that had already recovered P4M × 2::GFP but not PLC-PH::RFP. (**A′–C′**) Quantification of PM localizations of P4M × 2::GFP and PLC-PH::RFP (A′, n=18, 18, *Figure 1—source data 1*) or P4M × 2::GFP and Lgl::RFP (B′, n=20, 20, *Figure 1—source data 2*) or PLC-PH::GFP and Lgl::RFP (C′, n=20, 20, *Figure 1—source data 3*) during hypoxia and reoxygenation. PM Index: ratio of mean intensity of PM and cytosolic signals, normalized by the average value of the first three frames. Time stamp in *hr:min:sec* format. Scale bars: 5 μm.

The online version of this article includes the following video and source data for figure 1:

**Source data 1.** Hypoxia induces acute and reversible loss of PM PI4P and PIP2 in *Drosophila* follicle cells.

**Source data 2.** Hypoxia induces acute and reversible loss of PM PIP2 and Lgl in *Drosophila* follicle cells.

**Source data 3.** Hypoxia induces acute and reversible loss of PM PI4P and Lgl in *Drosophila* follicle cells.

**Figure 1—video 1.** Acute and reversible loss of PM localization of P4M × 2::GFP and PLC-PH::RFP under hypoxia in follicle cells.

https://elifesciences.org/articles/79582/figures#fig1video1

**Figure 1—video 2.** Acute and reversible loss of PM localization of PLC-PH::GFP and Lgl::mCherry under hypoxia in follicle cells.

https://elifesciences.org/articles/79582/figures#fig1video2

**Figure 1—video 3.** Acute and reversible loss of PM localization of P4M × 2::GFP and Lgl::mCherry under hypoxia in follicle cells.

https://elifesciences.org/articles/79582/figures#fig1video3

---

by flushing with normal air, both sensors rapidly recovered to the PM within ~10 min. At single-cell level, recovery of PM P4M × 2::GFP clearly and consistently preceded the PLC-PH::RFP (*Figure 1A*, *Figure 1—video 1*). Image quantification (see Materials and methods) further confirmed such differences in turnover dynamics between PM P4M × 2::GFP and PLC-PH::RFP (*Figure 1A′*). The faster depletion under hypoxia and delayed replenishment during reoxygenation of PM PIP2 suggest that PIP2 depletion likely involves its conversion to PI4P and its resynthesis depends on the recovery of PM PI4P.

In addition, under hypoxia the disappearance of P4M × 2::GFP intracellular puncta always preceded the depletion of PM P4M × 2::GFP. PM PI4P showed a transient increase at early phase of hypoxia

likely due to depletion of intracellular PI4P leading to increased amounts of free P4M × 2::GFP sensor that binds PM PI4P (*Figure 1A and A'*). Under reoxygenation, PM P4M × 2::GFP consistently recovered before the appearance of intracellular P4M × 2::GFP puncta, although the latter became brighter after recovery (*Figure 1A*, *Figure 1—video 1*). Such early depletion of intracellular PI4P pool under hypoxia and its delayed replenishment under reoxygenation suggest that cells appear to prioritize the maintenance of PM PI4P pool to the intracellular pool of PI4P under energetic stress such as hypoxia.

Finally, we investigated how electrostatic PM targeting of Lgl, a polybasic polarity protein carrying a typical polybasic motif (*Dong et al., 2015*), correlates with hypoxia-induced turnover of PM PI4P and PIP2. Quantitative live imaging of follicle epithelial cells expressing endogenous Lgl::RFP together with PLC-PH::GFP or P4M × 2::GFP showed that under hypoxia the loss of PM Lgl::RFP preceded PIP2 and PI4P, while under reoxygenation PM recovery of Lgl::RFP lagged behind both (*Figure 1B, C*, *Figure 1—videos 2; 3*). Such results are consistent with previous studies that electrostatic PM targeting of Lgl relies on both PIP2 and PI4P, although PIP2 appears to contribute more to the PM targeting of Lgl (*Dong et al., 2015*). Note that in *Figure 1B* Lgl::RFP recovery appears lower than in wild type, possibly due to the titration of PIP2 by PLC-PH::GFP expression.

Overall, our quantitatively live imaging data showed for the first time at high subcellular and temporal resolutions that hypoxia triggered a dramatic turnover of PM PI4P and PIP2 in vivo, which directly impacts the electrostatic PM targeting under hypoxia and reoxygenation.

## PI4KIIIα regulates the dynamic turnover of PM PI4P and PIP2 under hypoxia

The seven species of PPIns including PI4P and PIP2 are synthesized and interconverted by several dozens of PPIn kinases and phosphatases, many of which are conserved in *Drosophila* (*Balakrishnan*

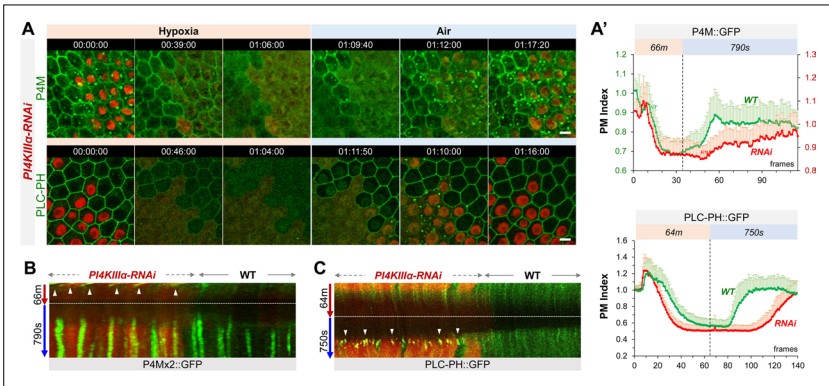

**Figure 2.** PI4KIIIα regulates PM PI4P and PIP2 homeostasis and dynamic turnover under hypoxia/reoxygenation. (**A**) Representative frames showing follicle cells expressing P4M::GFP and PLC-PH::GFP undergoing hypoxia and reoxygenation. *PI4KIIIα-RNAi* cells are labeled by RFP. (**A'**) PM localization of P4M::GFP (n=24, 23, *Figure 2—source data 1*) and PLC-PH::GFP (n=24, 24, *Figure 2—source data 2*) quantified in boundaries between wild type (WT) cells and between PI4KIIIα-RNAi (RNAi) cell. (**B**) Kymographs showing the persistent P4M::GFP puncta in both wild type and RNAi cells after hypoxia. White arrowheads point to puncta in RNAi cells at onset of hypoxia. Kymograph made by the maximum projection of 250 pixel wide line reslice of the time-lapse movie. (**C**) Kymograph showing the transient PLC-PH::GFP puncta (white arrowhead) in RNAi cells only. Kymograph made by the maximum projection of 300pixel wide line reslice of the time-lapse movie. Time stamp in *hr:min:sec* format. Scale bars: 5 µm.

The online version of this article includes the following video and source data for figure 2:

**Source data 1.** PI4KIIIα regulates PM PI4P homeostasis and dynamic turnover under hypoxia/reoxygenation.

**Source data 2.** PI4KIIIα regulates PM PIP2 homeostasis and dynamic turnover under hypoxia/reoxygenation.

**Figure 2—video 1.** PM P4M::GFP in *PI4KIIIα-RNAi* cells show accelerated loss under hypoxia and delayed recovery under reoxygenation.

https://elifesciences.org/articles/79582/figures#fig2video1

**Figure 2—video 2.** PM PLC-PH::GFP in *PI4KIIIα-RNAi* cells show accelerated loss under hypoxia and delayed recovery under reoxygenation.

https://elifesciences.org/articles/79582/figures#fig2video2

*et al., 2015*). We carried out a targeted RNAi screen to identify which PPIn kinases and phosphatases may be required for regulating the hypoxia-triggered dynamic turnover of PM PI4P and PIP2, using PM Lgl::GFP as a quick readout (*Supplementary file 1*). By imaging mosaic follicle epithelia containing both wild type and marked RNAi-expressing cells (*Figure 2A*), we eliminated the variability of each individual hypoxia imaging assay, making it possible to consistently and quantitatively detect even subtle phenotypes in RNAi cells. Among the candidates we identified is PtdIns-4 kinases IIIα (PI4KIIIα), one of the PI4K enzymes that phosphorylate PI to PI4P. Among them, PI4KIIIα is primarily responsible for the biosynthesis of PI4P in the PM (*Nakatsu et al., 2012*; *Tan et al., 2014*; *Yan et al., 2011*), while PI4KIIα and PI4KIIIβ (encoded by *four-wheel drive* or *fwd* in *Drosophila* [*Brill et al., 2000*]) are responsible for the synthesis of PI4P in endosomes and Golgi (*Baba et al., 2019*; *Burgess et al., 2012*; *Ketel et al., 2016*; *Tóth et al., 2006*).

Under normal (i.e. normoxia) conditions, *PI4KIIIα-RNAi* cells showed a moderate reduction of PM PI4P and increased intracellular PI4P puncta (*Figure 2A*). Under hypoxia, in both RNAi and wild-type cells PI4P intracellular puncta disappeared prior to the loss PM PI4P which showed similar depletion rates in two cell types (*Figure 2A*, *Figure 2—video 1*). Under reoxygenation, compared to wild-type cells, the recovery of PM PI4P in RNAi cells was significantly delayed while intracellular puncta showed much faster recovery (*Figure 2A and B*).

In contrast to P4M × 2::GFP, levels of PM PLC-PH::GFP in *PI4KIIIα-RNAi* cells were similar to the wild-type cells, suggesting a robust PM PIP2 homeostasis mechanism that compensates well the modest reduction of PI4P under normal conditions (consistent with [*Hammond and Burke, 2020*; *Sohn et al., 2018*]). However, once challenged by hypoxia, *PI4KIIIα-RNAi* cells showed much accelerated loss of PM PLC-PH::GFP (*Figure 2A, C*, *Figure 2—video 2*). Strikingly, under reoxygenation PLC-PH::GFP in *PI4KIIIα-RNAi* cells first formed transient but prominent intracellular puncta which were not seen in wild-type cells, and these puncta rapidly disappeared at the onset of PM PIP2 recovery which was strongly delayed compared to wild-type cells (*Figure 2A and C*, *Figure 2—video 2*).

In summary, our data support that PI4KIIIα is required for the efficient replenishment of PM PI4P and PIP2 after their hypoxia-triggered depletion. During reoxygenation, PI4KIIIα knock-down cells showed delayed PM PI4P recovery but enhanced replenishment of intracellular PI4P pool, although the latter could be due to increased amount of free P4M × 2::GFP sensors when PM P4P recovery was delayed. In addition, knocking down PI4KIIIα accelerates the depletion of PM PIP2 but not PI4P under hypoxia, suggesting an increased conversion of PIP2 to PI4P during depletion.

## PI4KIIα and FWD contribute to both PM and intracellular PI4P and PIP2 hemostasis and dynamic turnover

We then investigated how other two PI4K enzymes, PI4KIIα and FWD, contribute to the dynamic turnover of PI4P of PM and intracellular pools. Since neither *PI4KIIα* nor *fwd* null mutants are lethal (*Brill et al., 2000*; *Burgess et al., 2012*; *Polevoy et al., 2009*) and single RNAi knock-down against each showed no obvious phenotypes, we used newly published multi-RNAi tools *Qiao et al., 2018* to generate fly stocks simultaneously expressing multiple dsRNAs targeting either both *PI4KIIα* and *fwd* ('*PI4K-2KD*'), or all three PI4Ks ('*PI4K-3KD*'). While *PI4K-2KD* cells showed no discernable phenotypes under either normoxia or hypoxia in our hypoxia assays (*Figure 3—figure supplement 1*), *PI4K-3KD* cells showed severely reduced PM P4M × 2::GFP and dramatically increased intracellular P4M × 2::GFP puncta (*Figure 3A, B*); the latter could be due to more P4M × 2::GFP sensors being bound to the intracellular PI4P pool when PM PI4P is low (*Sohn et al., 2018*; *Wills et al., 2018*).

Similar to *PI4KIIIα-RNAi* cells, *PI4K-3KD* cells under hypoxia first showed reduction of intracellular P4M × 2::GFP puncta and transient increase of PM P4M × 2::GFP, followed by accelerated depletion of PM P4M × 2::GFP (*Figure 3A*, *Figure 3—video 1*). Under reoxygenation, *PI4K-3KD* cells only showed recovery of P4M × 2::GFP in intracellular puncta but not in PM, suggesting that *PI4K-3KD* cells are severely deficient in acute resynthesis of PM PI4P after its hypoxia-induced depletion.

Remarkably, PM PLC-PH::GFP levels showed no reduction in *PI4K-3KD* cells under normal conditions, despite of the severe loss of PM PI4P (*Figure 3A, B*, *Figure 3—video 2*). Similar to *PI4KIIIα-RNAi* cells, in *PI4K-3KD* cells PM PLC-PH::GFP showed accelerated loss under hypoxia (*Figure 3A*), and formed transient intracellular puncta during reoxygenation. Despite the apparent absence of PM PI4P recovery in *PI4K-3KD* cells, PM PLC-PH::GFP still recovered under reoxygenation, although the recovery was much delayed (*Figure 3A*, *Figure 3—video 2*).

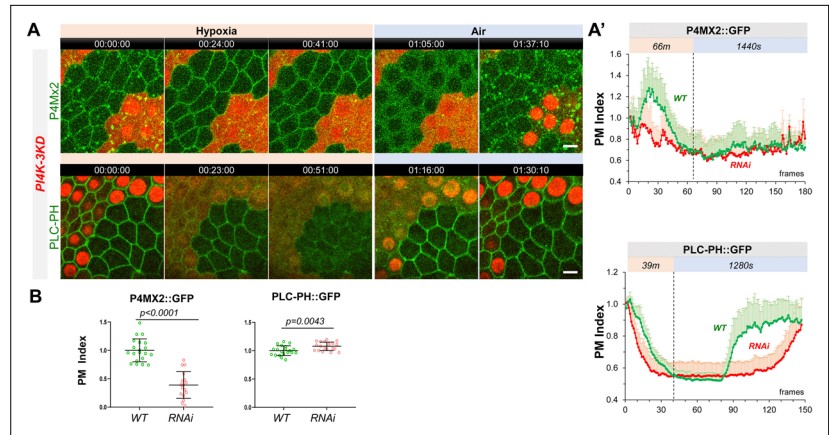

**Figure 3.** PM PI4P and PIP2 show accelerated loss under hypoxia and delayed recovery under reoxygenation in *PI4K-3KD* RNAi cells. (**A**) Representative frames showing follicle cells expressing P4M × 2::GFP or PLC-PH::GFP undergoing hypoxia and reoxygenation. *PI4K-3KD* cells are labeled by RFP. (**A'**) PM localization of P4M × 2::GFP (n=10, 5, *Figure 3—source data 1*) and PLC-PH::GFP (n=14, 15, *Figure 3—source data 2*) quantified in boundaries between wild type (WT) cells and between *PI4K-3KD* (RNAi) cell during hypoxia and regoxygenation. (**B**) Strong reduction of PM P4M × 2::GFP (n=20, 17, *Figure 3—source data 3*) but not PLC-PH::GFP (n=20, 20, *Figure 3—source data 4*) in *PI4K-3KD* follicle cells. Time stamp in hr:min:sec format. Scale bars: 5μm.

The online version of this article includes the following video, source data, and figure supplement(s) for figure 3:

**Source data 1.** PI4P showed accelerated loss under hypoxia and delayed recovery under reoxygenation in PI4K-3KD RNAi cells.

**Source data 2.** PIP2 showed accelerated loss under hypoxia and delayed recovery under reoxygenation in PI4K-3KD RNAi cells.

**Source data 3.** Reduction of PM PI4P in PI4K-3KD RNAi cells.

**Source data 4.** PM PI4P unchanged in PI4K-3KD RNAi cells.

**Source data 5.** Dynamic turnover of PM PI4P under hypoxia and reoxygenation in PI4K-2KD cells.

**Source data 6.** Dynamic turnover of PM PIP2 under hypoxia and reoxygenation in PI4K-2KD cells.

**Source data 7.** Dynamic turnover of PM Lgl under hypoxia and reoxygenation in PI4K-2KD cells.

**Figure supplement 1.** Dynamic turnover of PM P4M::GFP, PLC-PH::GFP and Lgl::GFP under hypoxia and reoxygenation in PI4K-2KD cells is similar to wild type cells.

**Figure 3—video 1.** PM P4M × 2::GFP in *PI4K-3KD-RNAi* cells show accelerated loss under hypoxia and delayed recovery under reoxygenation.
https://elifesciences.org/articles/79582/figures#fig3video1

**Figure 3—video 2.** PM PLC-PH::GFP in *PI4K-3KD-RNAi* cells show accelerated loss under hypoxia and delayed recovery under reoxygenation.
https://elifesciences.org/articles/79582/figures#fig3video2

---

Our data support that PI4KIIα and/or FWD contribute significantly to the maintenance of PM PI4P under normoxic conditions and to the replenishment of PM PI4P after hypoxia-triggered depletion. The data also suggest that an apparently PM PI4P-independent mechanism maintains the homeostatic level of PM PIP2 under normal conditions and sustains its recovery after hypoxia-triggered depletion. However, rapid replenishment of PM PI4P is clearly required for the efficient recovery of PM PIP2 after hypoxia-triggered depletion.

## PI4Ks regulate the electrostatic PM targeting and retargeting of Lgl::GFP

To investigate how electrostatic PM targeting is affected by the disruptions of PM PI4P and PIP2 turnover, we imaged the PM targeting of Lgl::GFP in *PI4KIIIα-RNAi* and *PI4K-3KD* cells undergoing hypoxia. In *PI4KIIIα-RNAi* cells, we found that Lgl::GFP essentially phenocopied the behavior of PLC-PH::GFP. As shown in *Figure 4A*, under normoxic conditions PM Lgl::GFP levels in *PI4KIIIα-RNAi* were similar to wild-type cells, but Lgl::GFP showed accelerated loss from PM under hypoxia and severely

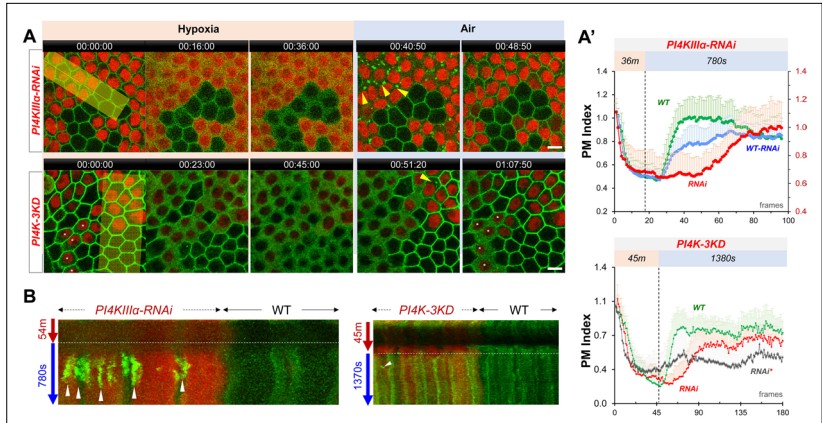

**Figure 4.** PI4Ks regulate the PM localization of Lgl::GFP under hypoxia and reoxygenation. (**A**) Representative frames showing Lgl::GFP PM localization during hypoxia and reoxygenation. RNAi cells are labeled by RFP. Yellow arrowheads: transient Lgl::GFP puncta (only few highlighted). *: RNAi cells that failed to recover Lgl::GFP to PM. (**A'**) (TOP) PM localization of Lgl::GFP quantified in boundaries between wild type (WT) cells (n=23), between *PI4KIIIα-RNAi* (RNAi) cells (n=24) and between WT and RNAi (WT-RNAi) cells (n=24). (*Figure 4—source data 1*). (BOTTOM). PM localization of Lgl::GFP quantified in boundaries between wild-type (WT) cells (n=20), between *PI4K-3KD* (RNAi) cells (n=10) and between cells failed recovery (RNAi*) (n=10). (*Figure 4—source data 2*). (**B**) Kymograph highlights the transient Lgl::GFP puncta (arrowheads). Each kymograph was made by reslicing the movie with the maximum projection of a 150 or 250-pixel wide line (yellow bands in A). Time stamp in *hr:min:sec* format. Scale bars: 5 μm.

The online version of this article includes the following video and source data for figure 4:

**Source data 1.** Dynamic turnover of PM Lgl under hypoxia and reoxygenation in PI4K-IIIα-RNAi cells.

**Source data 2.** Dynamic turnover of PM Lgl under hypoxia and reoxygenation in PI4K-3KD cells.

**Figure 4—video 1.** PM Lgl::GFP in *PI4KIIIα-RNAi* cells show accelerated loss under hypoxia and delayed recovery under reoxygenation.

https://elifesciences.org/articles/79582/figures#fig4video1

**Figure 4—video 2.** PM Lgl::GFP in *PI4K-3KD-RNAi* cells show accelerated loss under hypoxia and delayed recovery under reoxygenation.

https://elifesciences.org/articles/79582/figures#fig4video2

---

delayed recovery to PM during reoxygenation. In addition. Lgl::GFP also formed transient intracellular puncta prior to the onset of its PM recovery (*Figure 4A*, *Figure 4—video 1*). Such data are consistent with previous studies that electrostatic PM targeting of Lgl::GFP is more PIP2-dependent (*Dong et al., 2015*).

In *PI4K-3KD* cells, PM Lgl::GFP also showed accelerated loss under hypoxia and delayed recovery under reoxygenation (*Figure 4A*, *Figure 4—video 2*). Unlike PLC-PH::GFP, Lgl::GFP only formed few very transient puncta prior to the onset of PM recovery under reoxygenation (*Figure 4A*). However, in half (113/219) of *PI4K-3KD* cells PM Lgl::GFP was already partially diffused under normal conditions (asterisked in *Figure 4A*), and in these cells Lgl::GFP also failed to recover to PM during reoxygenation. Even in *PI4K-3KD* cells with normal PM Lgl::GFP, a third (36/106) failed to recover during reoxygenation. Such data suggest that the electrostatic PM targeting becomes much less resilient in PI4K-3KD cells, especially when cells are challenged by energetic stresses such as hypoxia.

## PM localization of PI4KIIIα is required for the dynamic turnover of PM PI4P and PIP2

Unlike PI4KIIα and FWD, which localize to intracellular membranes, PI4KIIIα is primarily PM localized (*Baskin et al., 2016*; *Nakatsu et al., 2012*). We thus investigated how subcellular localization to PI4KIIIα affects the hypoxia-induced dynamic turnover of PM PI4P. The PM localization of yeast PI4KIIIα ('Stt4p') requires EFR3, YPP1and Sfk1 (mammalian TTC7 and TMEM150A, respectively) (*Chung et al., 2015*; *Hammond et al., 2014*; *Nakatsu et al., 2012*). All three proteins are conserved in *Drosophila*, including EFR3 homologue Rbo ('Rolling blackout') (*Huang et al., 2004*; *Vijayakrishnan et al., 2009*),

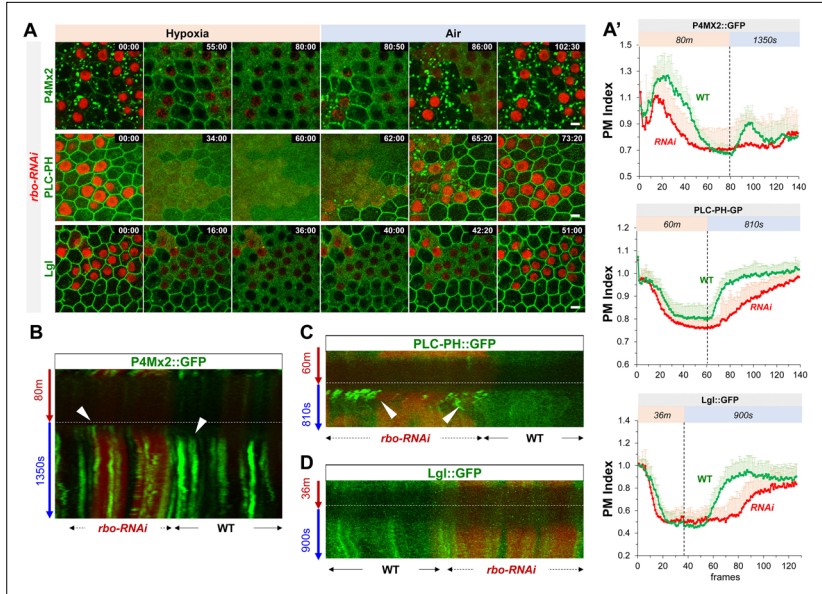

**Figure 5.** Rbo regulates the homeostasis and dynamic turnover of PI4P, PIP2 and Lgl under hypoxia/reoxygenation. (**A**) Representative frames follicle cells expressing P4M × 2::GFP or PLC-PH::GFP or Lgl::GFP undergoing hypoxia and reoxygenation. *rbo-RNAi* cells are labeled by RFP. Time stamp in *min:sec* format. (**A'**) PM localization of P4M::GFP (n=10, 10, *Figure 5—source data 1*), PLC-PH::GFP (n=20, 20, *Figure 5—source data 2*) and Lgl::GFP (n=20, 20, *Figure 5—source data 3*) in A quantified in wild type (WT) and *rbo-RNAi* (RNAi) cells. (**B**) Kymograph highlights the earlier onset of P4M::GFP puncta in post-hypoxia RNAi cells. White arrowheads point to the onset of puncta in post-hypoxia RNAi and WT cells. (**C**) Kymograph highlights the transient PLC-PH::GFP puncta (white arrowheads) seen only in post-hypoxia RNAi cells. (**D**) Kymograph showing the absence of Lgl::GFP puncta in post-hypoxia RNAi cells. Scale bars: 5μm.

The online version of this article includes the following video, source data, and figure supplement(s) for figure 5:

**Source data 1.** Dynamic turnover of PM PI4P under hypoxia and reoxygenation in rbo-RNAi cells.

**Source data 2.** Dynamic turnover of PM PIP2 under hypoxia and reoxygenation in rbo-RNAi cells.

**Source data 3.** Dynamic turnover of PM Lgl under hypoxia and reoxygenation in rbo-RNAi cells.

**Source data 4.** PM localization of Rbo is resistant to hypoxia.

**Figure supplement 1.** PM localization of Rbo is resistant to hypoxia.

**Figure 5—video 1.** PM localization of *rbo::GFP* is resistant to hypoxia.

https://elifesciences.org/articles/79582/figures#fig5video1

**Figure 5—video 2.** PM P4M × 2::GFP in *rbo-RNAi* cells show accelerated loss under hypoxia and delayed recovery under reoxygenation.

https://elifesciences.org/articles/79582/figures#fig5video2

**Figure 5—video 3.** PM PLC-PH::GFP in *rbo-RNAi* cells show accelerated loss under hypoxia and delayed recovery under reoxygenation.

https://elifesciences.org/articles/79582/figures#fig5video3

**Figure 5—video 4.** PM Lgl::GFP in Rbo-*RNAi* cells show accelerated loss under hypoxia and delayed recovery under reoxygenation.

https://elifesciences.org/articles/79582/figures#fig5video4

dYPP1/dTTC7 (CG8325) and dTMEM150A ("dTMEM", CG7990 and CG4025), (*Liu et al., 2018*). Consistent with that EFR3 is a peripheral membrane protein and that its PM targeting appears to be independent of PI4P/PIP2 (*Nakatsu et al., 2012*), in follicle epithelial cells Rbo::GFP (*Huang et al., 2004*; *Vijayakrishnan et al., 2009*) showed exclusive PM localization that was highly resistant to hypoxia (*Figure 5—figure supplement 1*, *Figure 5—video 1*). *rbo-RNAi* cells essentially phenocopied *PI4KIIIα-RNAi* cells in terms of the dynamic turnover of PM PI4P, PIP2 and Lgl::GFP under hypoxia, with one exception that under reoxygenation Lgl::GFP did not form transient puncta prior to PM recovery, even though PLC-PH::GFP still formed transient and prominent puncta in *rbo-RNAi* cells prior to the

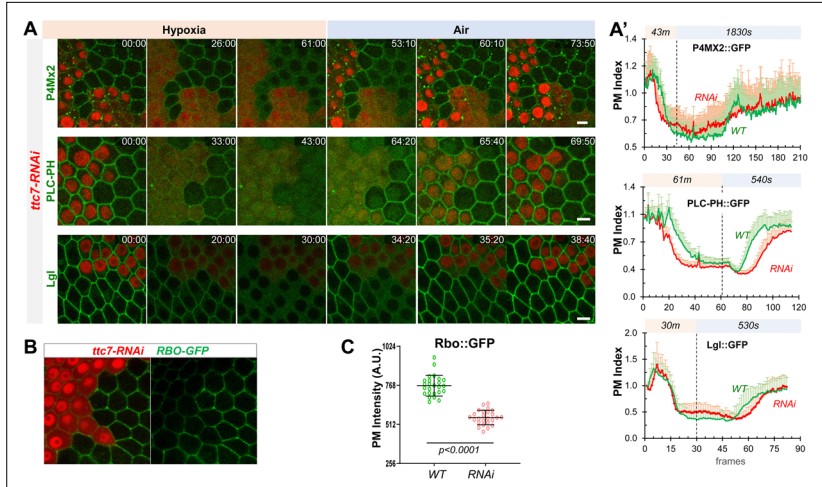

**Figure 6.** YPP1/TTC7 regulates the homeostasis and dynamic turnover of PI4P, PIP2 and Lgl in cells undergoing hypoxia and reoxygenation. (**A**) Representative frames follicle cells expressing P4M × 2::GFP or PLC-PH::GFP or Lgl::GFP undergoing hypoxia and reoxygenation. *ttc7-RNAi* cells are labeled by RFP. Time stamp in *min:sec* format. (**A′**) PM localization changes of P4M × 2::GFP (n=10, 10, *Figure 6—source data 1*), PLC-PH::GFP (n=20, 20, *Figure 6—source data 2*) and Lgl::GFP (n=20, 20, *Figure 6—source data 3*) in A quantified in wild type (WT) and *ttc7-RNAi* (RNAi) cells.(**B and C**) Reduction of PM RBO::GFP in *ttc7-1-RNAi* cells (n=24, 24, *Figure 6—source data 4*). Scale bars: 5μm.

The online version of this article includes the following video and source data for figure 6:

**Source data 1.** Dynamic turnover of PM PI4P under hypoxia and reoxygenation in ttc7-RNAi cells.

**Source data 2.** Dynamic turnover of PM PIP2 under hypoxia and reoxygenation in ttc7-RNAi cells.

**Source data 3.** Dynamic turnover of PM Lgl under hypoxia and reoxygenation in ttc7-RNAi cells.

**Source data 4.** Reduction of Rbo::GFP in ttc7-RNAi cells.

**Figure 6—video 1.** PM P4M × 2::GFP in *ttc7-RNAi* cells show loss under hypoxia and delayed recovery under reoxygenation.

https://elifesciences.org/articles/79582/figures#fig6video1

**Figure 6—video 2.** PM PLC-PH::GFP in *ttc7-RNAi* cells show accelerated loss under hypoxia and delayed recovery under reoxygenation.

https://elifesciences.org/articles/79582/figures#fig6video2

**Figure 6—video 3.** PM Lgl::GFP in *ttc7-RNAi* cells show accelerated loss under hypoxia and delayed recovery under reoxygenation.

https://elifesciences.org/articles/79582/figures#fig6video3

---

onset of the recovery of PM PLC-PH::GFP (*Figure 5A, C and D*, *Figure 5—videos 2–4*). The reason for such difference between Lgl::GFP and PLC-PH::GFP in *rbo-RNAi* cells is unclear.

YPP1/TTC7 helps to link PI4KIIIα to Rbo/EFR3 and enhances the PM targeting of PI4KIIIα in cultured cells (*Nakatsu et al., 2012*). Consistently, *ttc7-RNAi* cells showed similar albeit milder phenotypes in hypoxia-triggered turnover of PM PI4P and PIP2 as well as the PM targeting and retargeting of Lgl (*Figure 6A*, *Figure 6—videos 1–3*). Interestingly, *ttc7-RNAi* cells also showed reduced PM Rbo::GFP (*Figure 6B*), suggesting that TTC7 enhances the PM targeting of Rbo and that phenotypes in *ttc7-RNAi* cells could be partially due to the reduction of PM Rbo. Although at present we do not have the tools to directly examine the PI4KIIIα localization in *rbo-* or *ttc7-RNAi* cells, our data strongly support a scenario where PM localization of PI4KIIIα is essential for the efficient recovery of PM PI4P and PIP2 after hypoxia-triggered depletion.

## Acute ATP inhibition induces dynamic turnover of PM PI4P and PIP2 in HEK293 cells

How does hypoxia trigger the acute depletion of PM PI4P and PIP2? We showed previously that direct ATP inhibition by antimycin (AM) in follicle cells also induced loss of PM Lgl::GFP in follicle and

embryonic epithelial cells (*Dong et al., 2015*), suggesting that the acute depletion of PM PI4P and PIP2 could be triggered by hypoxia-induced ATP inhibition. We first tested this hypothesis using a genetically encoded ATP sensor AT[NL], which is a FRET-based ratiometric ATP sensor that recently became available and validated in *Drosophila* (*Imamura et al., 2009*; *Kioka et al., 2014*; *Tsuyama et al., 2017*). As shown in *Figure 7—figure supplement 1A*, live imaging of follicle cells expressing AT[NL] confirmed an acute and reversible reduction of intracellular ATP levels under hypoxia and reoxygenation. Because AT[NL] is not suitable for imaging together with our current PI4P and PIP2 sensors, we also tested an intensimetric ATP sensor MaLionR (*Arai et al., 2018*) in HEK293 cells. Under hypoxia, HEK293 cells showed acute and reversible reduction of ATP levels as measured by MaLionR intensity changes (*Figure 7—figure supplement 1B*, *Figure 7—video 1*), as well as reversible depletions of PM P4M × 2::GFP and PLC-PH::GFP which correlate with MaLionR intensity changes (*Figure 7—figure supplement 1C*).

We then subjected HEK293 cells to acute ATP inhibition by the treatment of 2-deoxyglucose (2-DG) and AM, followed by washout to allow ATP recovery. Upon adding 2-DG and AM, MaLionR brightness dropped rapidly within 10–20 min and plateaued afterward (*Figure 7A, B*, *Figure 7—videos 2; 3*). In general, P4M × 2::GFP or PLC-PH::GFP began gradually lost from PM at the onset of the ATP drop and became completely cytosolic within ~40–60 min of ATP inhibition (*Figure 7A, B*). While the reduction of ATP as measured by MaLionR brightness was rather uniform across cells, ATP recovery after the washout of 2-DG and AM was slightly asynchronous across the cells. In general, the recovery of PM PI4P always preceded the detectable increase of MaLionR brightness (*Figure 7A*, *Figure 7—video 2*), while PM PIP2 recovery was concurrent with MaLionR brightness increase (*Figure 7B*, *Figure 7—video 3*). In some cells PM PI4P recovery appeared up to ten minutes ahead of MaLionR increase (*Figure 7A*). Given that MaLionR appears to have a dynamic range between ~50 μM and 2 mM of ATP (*Arai et al., 2018*), such data support that initial replenishment of PM PI4P could start under very low intracellular ATP levels.

Similar to the results from hypoxia assays in *Drosophila* follicle cells (*Figure 1A*), in HEK293 cells under ATP inhibition the depletion of PM P4M × 2::GFP consistently lagged behind the loss of PM PLC-PH::GFP, while after drug washout PM P4M × 2::GFP recovery consistently preceded the recovery of PM PLC-PH::GFP (*Figure 7C*, *Figure 7—video 4*). We also investigated whether PI4KIIIα is required for the post-ATP inhibition recovery of PM PI4P in HEK293 cells. After ATP inhibition, we washed out drugs with medium containing 20 μM wortmannin (WM) which specifically inhibits PI4KIIIα/β isoforms but not PI4KIIα/β (*Balla and Balla, 2006*). Washout with WM caused no discernable delay in ATP recovery as measured by MaLionR brightness (*Figure 7D*), but strongly delayed PM recovery of PI4P, and to much less degree PIP2 (*Figure 7D*, *Figure 7—figure supplement 2*). Such results are consistent with the *Drosophila* RNAi results that PI4KIIIα is required for the efficient PM recovery of PI4P and PIP2 after hypoxia-triggered depletion.

In summary, ATP inhibition in human cultured cells also triggers acute and reversible depletion of PM PI4P and PIP2 similar to the dynamic turnover of PI4P and PIP2 in cells undergoing hypoxia. For reasons unknown, our efforts of making a transgenic MaLionR sensor in *Drosophila* was unsuccessful, limiting our ATP assays to HEK293 at present. However, our data strongly support that intracellular ATP levels directly dictate the homeostasis and turnover of PM PI4P and PIP2 in both *Drosophila* and human cultured cells.

## Discussion

### Dynamic turnover of PM PI4P and PIP2 trigged by hypoxia and ATP inhibition

We would speculate that the reduction of intracellular ATP levels, through either hypoxia or drug inhibition, triggers acute loss of PM PI4P and PIP2 by two possible mechanisms. PI4P and PIP2 could be maintained at slow turnover rates on the PM, but reduction of ATP activates a specific cellular response to acutely deplete PM PI4P and PIP2. Alternatively, a more parsimonious mechanism would be that PM PI4P and PIP2 are constantly under fast turnover, which requires high activity of PI and PIP kinases. ATP reduction, which directly inhibits the activity of these kinases, pivots the equilibrium to the dephosphorylation process which converts the PIP2 to PI4P and PI4P to PI.

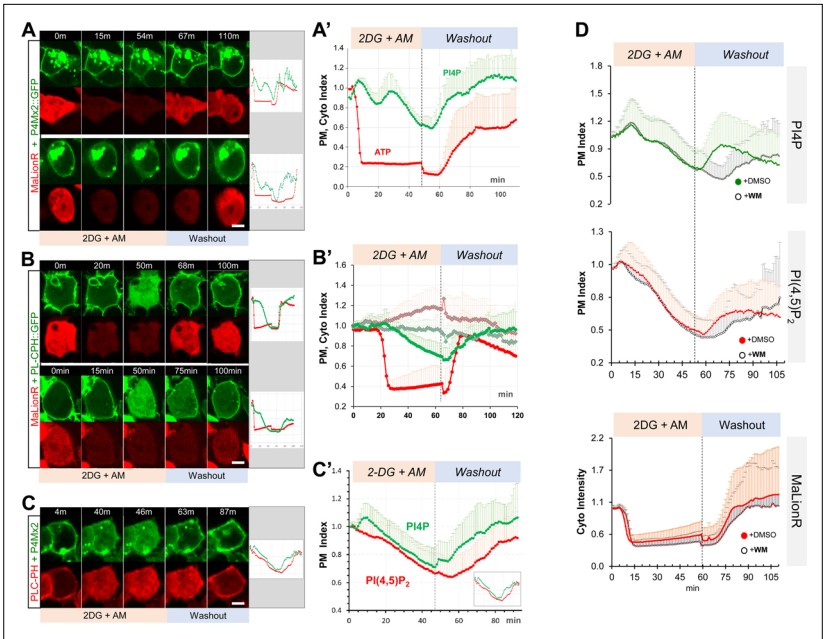

**Figure 7.** ATP inhibition induces acute and reversible loss of P4M × 2::GFP and PLC-PH::RFP from the PM in HEK293 cells. (**A, B**) Representative cells showing the PM localization of P4M × 2::GFP (**A**) or PLC-PH::GFP (**B**) and MaLionR ATP sensor during ATP inhibition and subsequent washout with low-glucose medium.(**A**) Top cell: P4M × 2::GFP recovery on PM was immediately followed by ATP sensor brightness increase (n=9). Bottom cell: measurable ATP increase lagged well behind the PM recovery of P4M × 2::GFP (n=6). Intracellular P4M × 2::GFP puncta were overexposed but excluded from quantification. (**A'**) Normalized quantification of PM localization of P4M × 2::GFP and MaLionR intensity (n=15, 19 cells, respectively). *Figure 7—source data 1*. (**B**) Top cell: synchronous PLC-PH::GFP recovery and ATP senor brightness increase (n=13). Bottom cell: PLC-PH::GFP PM recovery slightly preceded the detectable ATP increase (n=12). (**B'**) Normalized quantification of PM localization of PLC-PH::GFP (green) and MaLionR intensity (red) in ATP inhibition cells (solid dots, n=13) or serum-free medium treated cells (blank diamonds, n=12 cells) *Figure 7—source data 2*. (**C**) A representative cell showing the PM localization of P4M × 2::GFP and PLC-PH::RFP during ATP inhibition and subsequent washout. (**C'**) Quantification of PM P4M × 2::GFP and PLC-PH::GFP (n=13 cells) (*Figure 7—source data 3*). (**D**) Normalized PM localization index of P4M × 2::GFP, PLC-PH::RFP and cyto index of MaLionR sensor in cells treated with ATP inhibition followed by washout with buffer containing DMSO or Wortmannin (WM, 20 μm). (n=48, all samples). *Figure 7—source data 4; 5*. Scale bars: 10 μm.

The online version of this article includes the following video, source data, and figure supplement(s) for figure 7:

**Source data 1.** Correlation between ATP inhibition PM PI4P turnovery in HEK293 cells.

**Source data 2.** Correlation between ATP inhibition PM PIP2 turnovery in HEK293 cells.

**Source data 3.** ATP inhibition induces acute and reversible loss of PM PI4P and PIP2 in HEK293 cells.

**Source data 4.** Wortmannin inhibits PM PI4P recovery post ATP inhibition.

**Source data 5.** Post-inhibition recovery of ATP was not delayed by wortmannin treatment.

**Source data 6.** Acute and reversible reduction of intracellular ATP in *Drosophila* follicle cells undery hypoxia.

**Source data 7.** Acute and reversible reduction of intracellular ATP in HEK203 cells undery hypoxia.

**Figure supplement 1.** Real-time monitoring of intracellular ATP changes in follicular cells and HEK293 cells undergoing hypoxia and reoxygenation.

**Figure supplement 2.** Wortmannin inhibits PM PI4P recovery after ATP inhibition.

**Figure 7—video 1.** HEK293 cells expressing GFP and MaLionR ATP sensor undergoing hypoxia and reoxygenation.

https://elifesciences.org/articles/79582/figures#fig7video1

**Figure 7—video 2.** HEK293 cells expressing P4M × 2::GFP and MaLionR undergoing acute and transient ATP inhibition.

https://elifesciences.org/articles/79582/figures#fig7video2

*Figure 7 continued*

**Figure 7—video 3.** HEK293 cells expressing PLC-PH::GFP and MaLionR undergoing acute and transient ATP inhibition.

https://elifesciences.org/articles/79582/figures#fig7video3

**Figure 7—video 4.** HEK293 cells expressing P4M × 2::GFP and PLC-PH::RFP undergoing acute and transient ATP inhibition.

https://elifesciences.org/articles/79582/figures#fig7video4

Consistent with the critical role of PI4P in maintaining PM identity and its biological activity, our data revealed that cells undergoing hypoxia/ATP inhibition consistently prioritize the maintenance and recovery of PM PI4P over the intracellular PI4P pool in a PI4KIIIα-dependent manner. However, while PI4KIIIα is well characterized for its essential role in generating the PI4P on the PM (*Balla, 2013*; *Nakatsu et al., 2012*), $K_m$ATP values of PI4KIIIα (500–700 µM) and PI4KIIβ(Fwd) (~400 µM) are about one or two orders higher than that of PI4KIIα (10–50 µM) (*Balla and Balla, 2006*; *Carpenter and Cantley, 1990*). Such $K_m$ATP differences would suggest that, in contrast to our results, the intracellular PI4P pool should deplete more slowly and recover more quickly than the PM PI4P in cells undergoing hypoxia/ATP inhibition, as PI4KIIIα would be the first PI4K to lose activity under hypoxia and the last to become active under reoxygenation.

One possible reason behind such a discrepancy could be that $K_m$ATP of PI4KIIIα was measured decades ago using purified PI4KIIIα enzymes from tissues such as bovine brains and uterus (*Carpenter and Cantley, 1990*). Recent data showed that PI4KIIIα forms a highly ordered multi-protein membrane targeting complex essential for its activity (*Lees et al., 2017*). It is thus possible that PI4KIIIα in the complex may have much lower $K_m$ATP in vivo, and/or has dramatically increased enzymatic activity to produce sufficient PI4P at the PM even when ATP levels are much lower than the measured $K_m$. Alternatively, the $K_m$ATP of PI4KIIIα is indeed high and our live imaging results actually highlight a prioritized transfer of PI4P from the intracellular pool to maintain or replenish the PM PI4P levels. Phosphatidylinositol (PI) is abundant on intracellular membranes, but not the PM (*Pemberton et al., 2020*; *Zewe et al., 2020*). Therefore, during the early phase of reoxygenation when intracellular ATP levels are low, PI4P is first synthesized at the intracellular pool by PI4KIIα but is immediately transferred to the PM. Only after the full replenishment of PM PI4P is the intracellular PI4P pool filled. Supporting this transfer PI4P from intracellular pools to PM pools (*Dickson et al., 2014*), our data show loss of PM PI4P recovery in PI4K-3KD cells, in which the maintenance of intracellular pool of PI4P is supposedly impaired.

Out data are consistent with the view that in wild type cells under hypoxia/ATP inhibition, PM PIP2 depletes and recovers through direct inter-conversion with PI4P on the PM. Interestingly, in both *PI4KIIIα-RNAi* and *PI4K-3KD* cells, PM recovery of PIP2 is preceded with transient intracellular PIP2-positive puncta which were not seen in recovering wild-type cells. It is possible that in the absence or delay of PM PI4P recovery, enzymes such as PIP5K are instead electrostatically recruited to the intracellular PI4P-positive puncta (*Dong et al., 2016*; *Fairn et al., 2009*) to convert PI4P to PIP2. It is unclear, however, in PI4K knock down cells whether the delayed PM PIP2 recovery originates from the PIP2 generated in these puncta. Additional sensors are necessary to confirm the co-localization of PI4P and PIP2 on these transient puncta. Notably, MEF cells from PI4KIIIα knock-out mice also showed increased PIP2-positive intracellular vesicles (*Dong et al., 2016*; *Nakatsu et al., 2012*).

The existence of intracellular P4M × 2::GFP puncta in *PI4K-3KD* cells suggest that the knock down of *PI4KIIα* and *fwd* is unlikely complete, but the severe reduction of PM PI4P confirms the knock down is strong enough to greatly enhance the defects in *PI4KIIIα-RNAi* cells. Such partial knock-down by *PI4K-3KD* is actually necessary for our imaging assays, as completely blocking PI4P synthesis is cell lethal. It is striking that PM PIP2 is well maintained in the near absence of PM PI4P in PI4K-3KD cells. Synthetic biology-based evidence suggested that PIP5K can be sufficient to make PIP2 from PI in *E. coli* by phosphorylating both its fourth and fifth positions in the absence of PI4Ks (*Botero et al., 2019*), and it is possible that similar pathway maintains the steady state PM PIP2 levels in PI4K-3KD cells. Nonetheless, our imaging experiments showed that PI4K activity is essential for cells to maintain PM PIP2 levels when cells are subject to hypoxia. In this regard, our study of PI4K-compromised cells repeatedly revealed deficiencies in PI4P/PIP2 turnover and electrostatic PM targeting that can only be observed when cells are subject to energetic stress conditions.

## PM targeting of PI4KIIIα is crucial for maintaining and replenishing the PM PI4P

PM targeting of PI4KIIIα strictly depends on its formation of an obligate superassembly with TTC7 (YPP1), FAM126 (Hycin) and EFR3 (Rbo) (*Lees et al., 2017*; *Wu et al., 2014*). A recent study also showed that RNAi knock-downs of PI4KIIIα, TTC7 and Rbo yielded similar phenotypes in *Drosophila* wing discs, such as moderately reduced PM PI4P but no obvious changes of PM PIP2 (*Basu et al., 2020*). Same RNAi knock-downs in *Drosophila* photoreceptors also showed similar phenotypes such as reduced PI4P levels and impaired light response, although PIP2 levels also appear to be reduced (*Balakrishnan et al., 2018*). Moreover, our data showed that knocking down TTC7 also reduced PM localization of Rbo, supporting that components in PI4KIIIα complex may act interdependently for proper PM targeting in vivo.

The hypoxia-resistant PM localization of Rbo/dEFR3 suggests that under hypoxia/ATP inhibition PI4KIIIα maintains its PM localization, which should be essential for its role in recovering the PM PI4P. The core complex of PI4KIIIα/TTC7/FAM126 forms a collective basic surface that electrostatically binds to the acidic inner leaflet of the PM which could be sensitive to the loss of PM PI4P and PIP2. However, TTC7 also interacts with the C-terminus of EFR3 (*Chung et al., 2015*; *Lees et al., 2017*). PM targeting of yeast EFR3 requires a basic patch that interacts with general acidic phospholipids but is not disrupted by the loss of PM PI4P and PIP2 (*Wu et al., 2014*). Mammalian EFR3 contains an additional N-terminal Cys-rich palmitoylation site that is also required for the PM targeting (*Nakatsu et al., 2012*). Such dual and PI4P/PIP2-independent mechanisms are supported by the hypoxia-resistant PM localization of Rbo as we observed. Future studies will be needed to directly investigate the PM targeting of PI4KIIIα, TTC7 and FAM126 in vivo under hypoxia/ATP inhibition.

## Hypoxia/ATP inhibition-triggered PM PI4P and PIP2 turnover impacts the electrostatic targeting of polybasic proteins

While previous studies showed that genetically reducing PM PIP2 levels disrupts the PM localization of several polarity proteins including Lgl, it is difficult to conclude whether such loss of PM targeting is the direct consequence PIP2 reduction (*Claret et al., 2014*; *Gervais et al., 2008*). In this study, we are able to quantitatively and qualitatively demonstrate that in cells undergoing hypoxia-reoxygenation the acute and reversible loss of PM targeting of Lgl directly correlates with the turnover of PM PI4P and PIP2. Consistent with the idea that Lgl appears to depend more on PIP2 for its PM targeting, Lgl closely follows the dynamic turnover and relocation of PIP2 during hypoxia and reoxygenation. In particular, ectopic and transient puncta of Lgl::GFP seen in *PI4KIIIα-RNAi* or PI4K-3KD cells under reoxygenation appear to be strikingly similar to PIP2-positive puncta in these cells, although due to the limited array of biosensors we have not been able to directly confirm the co-localization of Lgl::GFP and PIP2 in these transient puncta. Additional genetically encoded biosensors for PI4P and PIP2 (e.g. P4M × 2::iRFP and PLC-PH::iRFP) are in development for such experiments.

It is notable that in *rbo-RNAi* cells, Lgl::GFP formed very few transient puncta prior to PM recovery during reoxygenation, even though PLC-PH::GFP showed plenty of prominent puncta. The reason for such difference between Lgl::GFP and PLC-PH::GFP in *rbo-RNAi* cells is unclear, though likely derives from the requirement of polybasic motif proteins for additional anionic lipids at the plasma membrane, specifically high molar fractions of phosphatidylserine in addition to lower concentrations of polyanionic phosphoinositides (*Yeung et al., 2008*).

In summary, our study revealed an acute and reversible loss of PI4P and PIP2 from PM under hypoxia/ATP inhibition in both *Drosophila* and human cultured cells. Such dynamic turnover of PM PI4P and PIP2 explains the dramatic loss of the PM targeting of polybasic polarity proteins such as Lgl under hypoxia. How cells meticulously maintain steady state PM PI4P and PIP2 levels under normal physiological conditions has been extensively studied; our studies highlight the importance of understanding mechanisms controlling this homeostasis and dynamics of phosphoinositides under energetic stresses triggered by hypoxia, ATP inhibition and ischemia, and the critical role of the interplay between polarity proteins and PM phosphoinositides in controlling cell polarity under normal and disease conditions.

# Materials and methods

## Key resources table

| Reagent type (species) or resource | Designation | Source or reference | Identifiers | Additional information |
|---|---|---|---|---|
| Genetic reagent (*D. melanogaster*) | *ubi-P4M::GFP* | This paper | | See "Materials and methods" |
| Genetic reagent (*D. melanogaster*) | *ubi-P4M × 2::GFP* | This paper | | See "Materials and methods" |
| Genetic reagent (*D. melanogaster*) | *ubi-PLC-PH::GFP* | This paper | | See "Materials and methods" |
| Genetic reagent (*D. melanogaster*) | *ubi-PLC-PH::RFP* | This paper | | See "Materials and methods" |
| Genetic reagent(*D. melanogaster*) | y[1] M{RFP[3xP3.PB] GFP[E.3xP3]=vas int. Dm}ZH-2A w[*]; PBac{y[+]-attP-9A} VK00022 | Bloomington *Drosophila* Stock Center | BDSC:24868; RRID:BDSC_24868 | |
| Genetic reagent (*D. melanogaster*) | y[1] w[*] P{y[+t7.7]=nos-phiC31\int.NLS}X; PBac{y[+]-attP-3B}VK00040 | Bloomington *Drosophila* Stock Center | BDSC:35568; RRID:BDSC_35568 | |
| Genetic reagent (*D. melanogaster*) | *PI4K-2KD* | This paper | | See "Materials and methods" |
| Genetic reagent (*D. melanogaster*) | *PI4K-3KD* | This paper | | See "Materials and methods" |
| Genetic reagent (*D. melanogaster*) | *UAS-AT1.03NL1* | DGRC#117,011 *Tsuyama et al., 2017* | DGRC#117,011 | |
| Genetic reagent (*D. melanogaster*) | *Cy2-Gal4* | Gift from David Bilder *Queenan et al., 1997* | | |
| Genetic reagent (*D. melanogaster*) | *rbo::GFP* | Gift from Kendal Broadie *Huang et al., 2004* | | |
| Genetic reagent (*D. melanogaster*) | *lgl::mCherry* | *Dong et al., 2015* | | |
| Genetic reagent (*D. melanogaster*) | *lgl::GFP* | *Dong et al., 2015* | | |
| Genetic reagent (*D. melanogaster*) | *UAS-PI4KIIIα-RNAi* | Bloomington *Drosophila* Stock Center | BDSC:35256; FLYB:FBst0035256; RRID:BDSC_35256 | FlyBase symbol: P{TRiP. GL00144}attP2 |
| Genetic reagent (*D. melanogaster*) | *UAS-Rbo-RNAi* | Vienna *Drosophila* Resource Center | VDRC:v47751; FLYB:Bst0467525 | FlyBase symbol: P{GD14013} v47751 |
| Genetic reagent (*D. melanogaster*) | *UAS-ttc7-RNAi* | Vienna *Drosophila* Resource Center | VDRC:v35881; FLYB: FBst0461391 | FlyBase symbol: P{GD13893} v35881 |
| Genetic reagent (*D. melanogaster*) | *UAS-Plc21C-RNAi* | Bloomington *Drosophila* Stock Center | BDSC:33719; FBst0033719 RRID:BDSC_33719 | |
| Genetic reagent (*D. melanogaster*) | *UAS-pis-RNAi* | Bloomington *Drosophila* Stock Center | BDSC:55602; FBst0055602; RRID:BDSC_55602 | |
| Genetic reagent (*D. melanogaster*) | *UAS-PI4KIIIα-RNAi* | Bloomington *Drosophila* Stock Center | BDSC:38242; FBst0038242; RRID:BDSC_38242 | |
| Genetic reagent (*D. melanogaster*) | *UAS-synj-RNAi* | Bloomington *Drosophila* Stock Center | BDSC:44420; FBst0044420; RRID:BDSC_44420 | |
| Genetic reagent (*D. melanogaster*) | *UAS-rdgβ-RNAi* | Bloomington *Drosophila* Stock Center | BDSC:44523; FBst0044523; RRID:BDSC_44523 | |
| Genetic reagent (*D. melanogaster*) | *UAS-INPP5E-RNAi* | Bloomington *Drosophila* Stock Center | BDSC:41701; FBst0041701; RRID:BDSC_41701 | |
| Genetic reagent (*D. melanogaster*) | *UAS-CG5026-RNAi* | Bloomington *Drosophila* Stock Center | BDSC:42759; FBst0042759; RRID:BDSC_42759 | |
| Genetic reagent (*D. melanogaster*) | *UAS-Pten-RNAi* | Bloomington *Drosophila* Stock Center | BDSC:33643; FBst0033643; RRID:BDSC_33643 | |
| Genetic reagent (*D. melanogaster*) | *UAS-Pi3k21B-RNAi* | Bloomington *Drosophila* Stock Center | BDSC:36810; FBst0036810; RRID:BDSC_36810 | |
| Genetic reagent (*D. melanogaster*) | *UAS-fwd-RNAi* | Bloomington *Drosophila* Stock Center | BDSC:35257; FBst0035257; RRID:BDSC_35257 | |

*Continued on next page*

*Continued*

| Reagent type (species) or resource | Designation | Source or reference | Identifiers | Additional information |
|---|---|---|---|---|
| Genetic reagent (*D. melanogaster*) | *UAS-mtm-RNAi* | Bloomington *Drosophila* Stock Center | BDSC:31552; FBst0031552; RRID:BDSC_31552 | |
| Genetic reagent (*D. melanogaster*) | *UAS-PIP4K-RNAi* | Bloomington *Drosophila* Stock Center | BDSC:35660; FBst0035660; RRID:BDSC_35660 | |
| Genetic reagent (*D. melanogaster*) | *UAS-PI4KIIIα-RNAi* | Bloomington *Drosophila* Stock Center | BDSC:35643; FBst0035643; RRID:BDSC_35643 | |
| Genetic reagent (*D. melanogaster*) | *UAS-INPP5E-RNAi* | Bloomington *Drosophila* Stock Center | BDSC:34037; FBst0034037; RRID:BDSC_34037 | |
| Genetic reagent (*D. melanogaster*) | *UAS-CG3632-RNAi* | Bloomington *Drosophila* Stock Center | BDSC:38341; FBst0038341; RRID:BDSC_38341 | |
| Genetic reagent (*D. melanogaster*) | *UAS-mtm-RNAi* | Bloomington *Drosophila* Stock Center | BDSC:38339; FBst0038339; RRID:BDSC_38339 | |
| Genetic reagent (*D. melanogaster*) | *UAS-PIP4K-RNAi* | Bloomington *Drosophila* Stock Center | BDSC:35338; FBst0035338; RRID:BDSC_35338 | |
| Genetic reagent (*D. melanogaster*) | *UAS-INPP5E-RNAi* | Bloomington *Drosophila* Stock Center | BDSC:34037; FBst0034037; RRID:BDSC_34037 | |
| Genetic reagent (*D. melanogaster*) | *UAS-CG3632-RNAi* | Bloomington *Drosophila* Stock Center | BDSC:38341; FBst0038341; RRID:BDSC_38341 | |
| Genetic reagent (*D. melanogaster*) | *UAS-mtm-RNAi* | Bloomington *Drosophila* Stock Center | BDSC:38339; FBst0038339; RRID:BDSC_38339 | |
| Genetic reagent (*D. melanogaster*) | *UAS-PIP4K-RNAi* | Bloomington *Drosophila* Stock Center | BDSC:35338; FBst0035338; RRID:BDSC_35338 | |
| Genetic reagent (*D. melanogaster*) | *UAS-FIG4-RNAi* | Bloomington *Drosophila* Stock Center | BDSC:38291; FBst0038291; RRID:BDSC_38291 | |
| Genetic reagent (*D. melanogaster*) | *UAS-CG5026-RNAi* | Bloomington *Drosophila* Stock Center | BDSC:38309; FBst0038309; RRID:BDSC_38309 | |
| Genetic reagent (*D. melanogaster*) | *UAS-PI4KIIα-RNAi* | Bloomington *Drosophila* Stock Center | BDSC:35278; FBst0035278; RRID:BDSC_35278 | |
| Genetic reagent (*D. melanogaster*) | *UAS-norpA-RNAi* | Bloomington *Drosophila* Stock Center | BDSC:31113; FBst0031113; RRID:BDSC_31113 | |
| Genetic reagent (*D. melanogaster*) | *UAS-sktl-RNAi* | Bloomington *Drosophila* Stock Center | BDSC:27715; FBst0027715; RRID:BDSC_27715 | |
| Genetic reagent (*D. melanogaster*) | *UAS-fwd-RNAi* | Bloomington *Drosophila* Stock Center | BDSC:29396; FBst0029396; RRID:BDSC_29396 | |
| Genetic reagent (*D. melanogaster*) | *UAS-CG6707-RNAi* | Bloomington *Drosophila* Stock Center | BDSC:28316; FBst0028316; RRID:BDSC_28316 | |
| Genetic reagent (*D. melanogaster*) | *UAS-synj-RNAi* | Bloomington *Drosophila* Stock Center | BDSC:34378; FBst0034378; RRID:BDSC_34378 | |
| Genetic reagent (*D. melanogaster*) | *UAS-pis-RNAi* | Bloomington *Drosophila* Stock Center | BDSC:29383; FBst0029383; RRID:BDSC_29383 | |
| Genetic reagent (*D. melanogaster*) | *UAS-sl-RNAi* | Bloomington *Drosophila* Stock Center | BDSC:32385; FBst0032385; RRID:BDSC_32385 | |
| Genetic reagent (*D. melanogaster*) | *UAS-CG42271-RNAi* | Bloomington *Drosophila* Stock Center | BDSC:29411; FBst0029411; RRID:BDSC_29411 | |
| Genetic reagent (*D. melanogaster*) | *UAS-Plc21C-RNAi* | Bloomington *Drosophila* Stock Center | BDSC:31270; FBst0031270; RRID:BDSC_31270 | |
| Genetic reagent (*D. melanogaster*) | *UAS-Pi3K92E-RNAi* | Bloomington *Drosophila* Stock Center | BDSC:35798; FBst0035798; RRID:BDSC_35798 | |
| Genetic reagent (*D. melanogaster*) | *UAS-Pi3K59F-RNAi* | Bloomington *Drosophila* Stock Center | BDSC:33384; FBst0033384; RRID:BDSC_33384 | |
| Genetic reagent (*D. melanogaster*) | *UAS-sl-RNAi* | Bloomington *Drosophila* Stock Center | BDSC:35604; FBst0035604; RRID:BDSC_35604 | |

*Continued on next page*

*Continued*

| Reagent type (species) or resource | Designation | Source or reference | Identifiers | Additional information |
|---|---|---|---|---|
| Genetic reagent (*D. melanogaster*) | UAS-CG9784-RNAi | Bloomington *Drosophila* Stock Center | BDSC:34723; FBst0034723; RRID:BDSC_34723 | |
| Genetic reagent (*D. melanogaster*) | UAS-Pi3K68D-RNAi | Bloomington *Drosophila* Stock Center | BDSC:34621; FBst0034621; RRID:BDSC_34621 | |
| Genetic reagent (*D. melanogaster*) | UAS-Ocrl-RNAi | Bloomington *Drosophila* Stock Center | BDSC:34722; FBst0034722; RRID:BDSC_34722 | |
| Genetic reagent (*D. melanogaster*) | UAS-sktl-RNAi | Bloomington *Drosophila* Stock Center | BDSC:35198; FBst0035198; RRID:BDSC_35198 | |
| Genetic reagent (*D. melanogaster*) | UAS-Pten-RNAi | Bloomington *Drosophila* Stock Center | BDSC:25841; FBst0025841; RRID:BDSC_25841 | |
| Genetic reagent (*D. melanogaster*) | UAS-CG33981,fab1-RNAi | Bloomington *Drosophila* Stock Center | BDSC:35793; FBst0035793; RRID:BDSC_35793 | |
| Genetic reagent (*D. melanogaster*) | UAS-Plc21C-RNAi | Bloomington *Drosophila* Stock Center | BDSC:32438; FBst0032438; RRID:BDSC_32438 | |
| Genetic reagent (*D. melanogaster*) | UAS-rdgB-RNAi | Bloomington *Drosophila* Stock Center | BDSC:28796; FBst0028796; RRID:BDSC_28796 | |
| Genetic reagent (*D. melanogaster*) | UAS-CG3530-RNAi | Bloomington *Drosophila* Stock Center | BDSC:25864; FBst0025864; RRID:BDSC_25864 | |
| Genetic reagent (*D. melanogaster*) | UAS-Synj-RNAi | Bloomington *Drosophila* Stock Center | BDSC:27489; FBst0027489; RRID:BDSC_27489 | |
| Genetic reagent (*D. melanogaster*) | UAS-Pi3K68D-RNAi | Bloomington *Drosophila* Stock Center | BDSC:31252; FBst0031252; RRID:BDSC_31252 | |
| Genetic reagent (*D. melanogaster*) | UAS-Pten-RNAi | Bloomington *Drosophila* Stock Center | BDSC:25967; FBst0025967; RRID:BDSC_25967 | |
| Genetic reagent (*D. melanogaster*) | UAS-Pi3K92E-RNAi | Bloomington *Drosophila* Stock Center | BDSC:27690; FBst0027690; RRID:BDSC_27690 | |
| Genetic reagent (*D. melanogaster*) | UAS-norpA-RNAi | Bloomington *Drosophila* Stock Center | BDSC:31197; FBst0031197; RRID:BDSC_31197 | |
| Genetic reagent (*D. melanogaster*) | UAS-fwd-RNAi | Bloomington *Drosophila* Stock Center | BDSC:31187; FBst0031187; RRID:BDSC_31187 | |
| Genetic reagent (*D. melanogaster*) | UAS-Plc21C-RNAi | Bloomington *Drosophila* Stock Center | BDSC:31269; FBst0031269; RRID:BDSC_31269 | |
| Genetic reagent (*D. melanogaster*) | UAS-sl-RNAi | Bloomington *Drosophila* Stock Center | BDSC:32906; FBst0032906; RRID:BDSC_32906 | |
| Genetic reagent (*D. melanogaster*) | UAS-CG3530-RNAi | Bloomington *Drosophila* Stock Center | BDSC:38340; FBst0038340; RRID:BDSC_38340 | |
| Genetic reagent (*D. melanogaster*) | UAS-CG5026-RNAi | Bloomington *Drosophila* Stock Center | BDSC:57020; FBst0057020; RRID:BDSC_57020 | |
| Genetic reagent (*D. melanogaster*) | UAS-PIP4K-RNAi | Bloomington *Drosophila* Stock Center | BDSC:65891; FBst0065891; RRID:BDSC_65891 | |
| Genetic reagent (*D. melanogaster*) | UAS-Sac1-RNAi | Bloomington *Drosophila* Stock Center | BDSC:56013; FBst0056013; RRID:BDSC_56013 | |
| Genetic reagent (*D. melanogaster*) | UAS-Pi3K59F-RNAi | Bloomington *Drosophila* Stock Center | BDSC:64011; FBst0064011; RRID:BDSC_64011 | |
| Genetic reagent (*D. melanogaster*) | UAS-Pi3K68D-RNAi | Bloomington *Drosophila* Stock Center | BDSC:35265; FBst0035265; RRID:BDSC_35265 | |
| Genetic reagent (*D. melanogaster*) | UAS-PIP5K59B-RNAi | Bloomington *Drosophila* Stock Center | BDSC:62855; FBst0062855; RRID:BDSC_62855 | |
| Genetic reagent (*D. melanogaster*) | UAS-Pi3K93E-RNAi | Bloomington *Drosophila* Stock Center | BDSC:61182; FBst0061182; RRID:BDSC_61182 | |
| Genetic reagent (*D. melanogaster*) | UAS-mtm-RNAi | Bloomington *Drosophila* Stock Center | BDSC:57298; FBst0057298; RRID:BDSC_57298 | |

*Continued*

| Reagent type (species) or resource | Designation | Source or reference | Identifiers | Additional information |
|---|---|---|---|---|
| Genetic reagent (*D. melanogaster*) | *UAS-Pi3K21B-RNAi* | Bloomington *Drosophila* Stock Center | BDSC:38991; FBst0038991; RRID:BDSC_38991 | |
| Genetic reagent (*D. melanogaster*) | *UAS-Pi3K59F-RNAi* | Bloomington *Drosophila* Stock Center | BDSC:36056; FBst0036056; RRID:BDSC_36056 | |
| Genetic reagent (*D. melanogaster*) | *UAS-FIG4-RNAi* | Bloomington *Drosophila* Stock Center | BDSC:58063; FBst0058063; RRID:BDSC_5806335265 | |
| Cell line (*Homo-sapiens*) | HEK293 | ATCC | CRL-1573 | |
| Recombinant DNA reagent | P4M::GFP (plasmid) | *Hammond et al., 2014* | | |
| Recombinant DNA reagent | P4M × 2::GFP (plasmid) | *Hammond et al., 2014* | | |
| Recombinant DNA reagent | PLC-PH::GFP (plasmid) | *Hammond et al., 2014* | | |
| Recombinant DNA reagent | pGU (plasmid) | *Lu et al., 2021* | | |
| Recombinant DNA reagent | MaLionR (plasmid) | Addgene | Addgene #113908 | |
| Recombinant DNA reagent | pNP (plasmid) | *Qiao et al., 2018* | | |
| Recombinant DNA reagent | pNP-fwd-KD (plasmid) | This paper | | |
| Recombinant DNA reagent | pNP-PI4KII-KD (plasmid) | This paper | | |
| Recombinant DNA reagent | pNP-PI4KIIIα-KD (plasmid) | This paper | | |
| Recombinant DNA reagent | pNP-PI4K-2KD (plasmid) | This paper | | |
| Recombinant DNA reagent | pNP-PI4K-3KD (plasmid) | This paper | | |
| Commercial assay or kit | Plasmid midi kid | Qiagen | Cat#12,143 | |
| Commercial assay or kit | Plasmid mini kit | Thermo Scientific | Cat#K0503 | |
| Commercial assay or kit | Gel extraction | Thermo Scientific | Cat#K0692 | |
| Chemical compound, drug | Halocarbon oil | Halocarbon 95 oil | Cat#9002-83-9 | |
| Chemical compound, drug | X-treme Gene 9 DNA transfection reagent | Sigma | Cat#6365787001 | |
| Chemical compound, drug | Fluorescent PM dye | ThermoFisher | ThermoFisher, Cat#C10046 | |
| Chemical compound, drug | 2-Deoxy-D-glucose | Sigma | Cat#D8375 | |
| Chemical compound, drug | Antimycin A | Sigma | Cat#A8674 | |
| Chemical compound, drug | Membrane dye | Invitrogen CellMask | Cat# C10046 | |
| Chemical compound, drug | Wortmannin | Sigma | Cat# 19545-26-7 | |
| Software, algorithm | Fiji (ImageJ) | https://imagej.net/Fiji | https://imagej.net/Fiji | |
| Software, algorithm | GraphPad Prism 8.0 | GraphPad Software | http://www.graphpad.com/ | |
| Other | air-permeable membrane | YSI Inc | YSI Membrane Model #5,793 | YSI Inc, Yellow Springs, OH |

## Fly stocks

Flies of carrying transgenic *ubi-P4M::GFP, ubi-P4M × 2::GFP, ubi-PLC-PH::GFP and ubi-PLC-PH::RFP* alleles were generated by *phiC31*-mediated integration protocol (*Huang et al., 2009*). *attP$^{VK00022}$* (BL#24868) and *attP$^{VK00040}$* (BL#35568) stocks were used to integrate the above constructs to the 2nd chromosome and 3rd chromosome, respectively.

PI4K-3KD was generated using pNP plasmid based on the published protocol *Qiao et al., 2018*. The pNP vector was cut with Nhe I and EcoR I and ligated with annealed oligo-dimmer o to generate pNP-fwd-KD, pNP-PI4KII-KD, and pNP-PI4KIIIα-KD. Primers used for annealing oligo-dimmer were listed in *Supplementary file 2*:

The pNP-fwd-KD was digested with Spe I and ligated with the shortest fragment cut by Spe I and Xba I from pNP-PI4KIIα-KD to make pNP-PI4K-2KD. To make pNP-PI4K-3KD, pNP-PI4KIIIα-KD was digested with Spe I and ligated with the shortest fragment cut by Spe I and Xba I from pNP-PI4K-2KD. pNP-PI4K-2KD and pNP-PI4K-3KD were used to generate transgenic stocks with the standard protocol.

UAS-AT1.03NL1(DGRC#117011) was used to express the *Drosophila*-optimized ATeam ATP sensor 'AT[NL]' in follicle cells (*Tsuyama et al., 2017*). cy2-Gal4 (*Queenan et al., 1997*) was a gift from David Bilder, UC Berkeley. *rbo::GFP* was a gift from Kendal Broadie, Vanderbilt University (*Huang et al., 2004*).

*w; lgl::mCherry* and *w; lgl::GFP* knock-in stock were previously published (*Dong et al., 2015*). *Drosophila* cultures and genetic crosses are carried out at 25 °C.

Additional stocks used were: *UAS-PI4KIIIα-RNAi* (BL#35256), *UAS-Rbo-RNAi* (VDRC#47751), *UAS-ttc7-RNAi* (VDRC#35881).

## Molecular cloning

Mammalian constructs of P4M::GFP, P4M × 2::GFP, and PLC-PH::GFP were as previously described (*Hammond et al., 2014*; *Hammond et al., 2012*; *Várnai and Balla, 1998*). DNA fragments encoding PLC-PH::GFP, P4M::GFP and P4M × 2::GFP were inserted into pGU vector (*Lu et al., 2021*) which contains a ubiquitin promoter. MaLionR ATP sensor was obtained from Addgene (#113908) (*Arai et al., 2018*).

## Generation of RNAi mutant clones in *Drosophila* follicle epithelia

Follicle cells containing over-expressing or RNAi clones were generated by heat-shocking 3 days old (after eclosion) young females of the correct genotype at 37 °C for 15–30 min and ovaries were dissected 3 days later.

## Live imaging and hypoxia treatment in *Drosophila* epithelial cells

Ovaries from adult females of 2 days old were dissected in halocarbon oil (#95) and were imaged according to previously published protocol (*Dong et al., 2015*; *Huang et al., 2011*). To ensure sufficient air exchange to samples during the imaging session, dissected ovaries were mounted in halocarbon oil on an air-permeable membrane (YSI Membrane Model #5793, YSI Inc, Yellow Springs, OH) sealed by vacuum grease on a custom-made plastic slide over a 10 × 10 mm$^2$ cut-through window. The slide was then mounted in a custom made air-tight micro chamber (~5 cm$^3$) for live imaging under confocal microscope. Oxygen levels inside the chamber were controlled by flow of either air or custom $O_2/N_2$ gas mixture at the rate of approximately 1–5 cc/s. Images were captured at room temperature (25 °C) on an Olympus FV1000 confocal microscope (60 x Uplan FL N oil objective, NA = 1.3) by Olympus FV10-ASW software, or on a Nikon A1 confocal microscope (Plan Fluo 60 x oil objective, NA = 1.3) by NIS-Elements AR software.

## Cell culture and imaging

HEK293 cells were cultured in glass bottom dishes (In Vitro Scientific) and were transfected with DNA using X-treme Gene 9 DNA transfection reagent (Sigma Cat# 6365787001). After 24–40 hr of transfection cells were mounted and imaged on a Nikon A1 confocal microscope (Plan Fluo 40 x oil objective, NA = 1.3) by NIS-Elements AR software. For images to be used for quantification, parameters were carefully adjusted to ensure no or minimum overexposure. In addition, when necessary, a fluorescent PM dye (CellMask DeepRed Plasma Membrane Stain, ThermoFisher, Cat#C10046) was added to the cell culture prior to live imaging to help in visualizing the PM for later quantifications.

## Hypoxia and ATP inhibition experiments in HEK293 cells

HEK293 cells one day after transfection were imaged live in temperature control chamber at 37 °C. For hypoxia and ATP inhibition experiments, cells were starved in glucose-free medium six hours prior to live imaging. Hypoxia treatment was carried out using a custom designed culture dish lid which seals the 35-mm glass-bottom culture dish but allows gas to be flushed in and out the sealed dish chamber through two small built-in nozzles. Prior to the sealing, medium inside dish was reduced to ~200 μl and was covered by an air permeable membrane to prevent evaporation. Oxygen levels inside the chamber were controlled by flow of either air or custom $O_2/N_2$ gas mixture at the rate of approximately 0.1 cc/s. ATP inhibition was initiated by adding equal volume of serum containing 2-DG and antimycin to the final concentrations of 10 mM and 2 μM, respectively. After the end of inhibition, drugs were washout by replacing with normal media. For wortmannin inhibition experiment, wortmannin was added to the washout media to final concentration of 10 μM.

## Image processing and quantification

Time-lapse movies were first stabilized by HyperStackReg plug-in in ImageJ. Images or movies containing excessively noisy channels were denoised by PureDenoise plugin in ImageJ prior to quantification. PM localization of GFP or RFP in images or movies was measured in Image J by custom macro scripts. For *Drosophila* samples, ROIs approximately 20–40 μm² were drawn across selected cell junctions in the first frame of the movie. Custom macros were used to automatically generate PM masks by threshold-segmentation using the mean pixel value of the ROI.

For HEK293 cells, PM masks were generated by an à trous waveleta decomposition method (*Hammond et al., 2014*; *Olivo-Marin, 2002*) base on the channel that either contains PM-localized proteins or fluorescent PM dyes. Cytosol masks were generated by segmentation using threshold based on the mean pixel value of the ROI. Cells expressing all transfected fluorescent proteins were selected for measurement. For each cell, one ROI was drawn to cover part of cell that contains PM and cytoplasm.

Due to the use of computer generated PM and cytosol masks, the exact shape of the ROI was not critical, except that PM segments in contact with neighboring expressing cells were avoided. Nuclei and intracellular puncta were also avoided. Custom macros were used to automatically measure PM and cytosolic intensities of each fluorescent protein in each cell marked by ROI in the sample image. Background was auto-detected by the macro based on the minimal pixel value of the whole image.

The PM localization index for each fluorescent protein was auto-calculated by the macro as the ratio of [PM - background]/[cytosol - background]. In live imaging experiments, "Normalized PM Index" was calculated by normalizing (PM Index –1) over the period of recording against the (PM Index –1) at 0 min. Data were further processed in Excel, visualized and analyzed in Graphpad Prism.

## Acknowledgements

We are grateful to Drs Kendal Broadie, David Bilder and Tadashi Uemura for reagents and fly stocks, Kriti Sanghi for technical assistances, anonymous reviewers for their helpful comments, Dr Simon Watkins and University of Pittsburgh Medical School Center for Biologic Imaging for generous imaging and microscopy support, Bloomington and Kyoto Stock Centers for fly stocks, and Developmental Studies Hybridoma Bank (DSHB) for antibodies. Funding: This work was supported by grants NIH- NCRR R21RR024869 (Y H), NIH-NIGMS R01GM086423 and R01GM121534 (Y H), NIH 1R35GM119412-01 (G R H). University of Pittsburgh Medical School Center for Biologic Imaging is supported by grant 1S10OD019973-01 from NIH.

## Additional information

### Funding

| Funder | Grant reference number | Author |
| --- | --- | --- |
| National Institute of General Medical Sciences | R01GM121534 | Yang Hong |
| National Institute of General Medical Sciences | R01GM086423 | Yang Hong |
| National Institute of General Medical Sciences | R35GM119412 | Gerald R Hammond |
| National Institute of General Medical Sciences | R21RR024869 | Yang Hong |

The funders had no role in study design, data collection and interpretation, or the decision to submit the work for publication.

### Author contributions

Juan Lu, Data curation, Formal analysis, Investigation, Methodology, Validation, Visualization, Writing – review and editing; Wei Dong, Data curation, Formal analysis, Investigation, Project administration, Validation, Visualization, Writing – review and editing; Gerald R Hammond,

Conceptualization, Formal analysis, Funding acquisition, Methodology, Software, Supervision, Writing – original draft, Writing – review and editing; Yang Hong, Conceptualization, Data curation, Formal analysis, Funding acquisition, Investigation, Methodology, Project administration, Resources, Software, Supervision, Validation, Visualization, Writing – original draft, Writing – review and editing

## Author ORCIDs
Gerald R Hammond ![ORCID] http://orcid.org/0000-0002-6660-3272
Yang Hong ![ORCID] http://orcid.org/0000-0003-2252-0798

## Decision letter and Author response
Decision letter https://doi.org/10.7554/eLife.79582.sa1
Author response https://doi.org/10.7554/eLife.79582.sa2

## Additional files

### Supplementary files
• MDAR checklist

• Supplementary file 1. Candidate RNAi screen on regulators controlling hypoxia-induced acute and reversible loss of PM Lgl.

• Supplementary file 2. Primers used in this study.

### Data availability
All data generated or analysed during this study are included in the manuscript and supporting files.

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
