## [Editor Report]

The authors show that hypoxia leads to previously unappreciated effects on levels of plasma membrane PI4P and PIP2, which affects membrane targeting of proteins important for normal cellular physiology, including cell polarity. They provide insight into the identity of the PI4Ks that are responsible for regenerating plasma membrane PIP2 following return to normoxia. These findings are novel and provide multiple interesting insights for those studying phosphoinositide biology as well as cellular responses to hypoxic stress and recovery.

---

## [Decision Letter]

[Editors' note: this paper was reviewed by Review Commons.]

---

## [Author Response]

We would like to thank all three reviewers for their comprehensive and constructive comments. We are in particular grateful to reviewer#3 for the long and detailed suggestions on improving the manuscript text and figures. We have already incorporated most of these suggestions into the preliminary revision of manuscript. In addition to the changes already made in initial submission to *eLife*, in this submission:

1. We have successfully carried out hypoxia assays in live HEK293 cells as requested by the reviewers. Such results are presented in updated Figure 7—figure supplement 1B&C. In Brief, we showed that hypoxia in HEK293 cells induced acute and reversible reduction of intracellular ATP as measured in real time by ATP sensor MaLionR, as well as depletion of PM PI4P and PIP2. Such results are fully consistent with our hypoxia assays in *Drosophila* tissues, further supporting the hypothesis that hypoxia triggers acute and reversible depletion of PM PI4P and PIP2 through ATP inhibition. We also added a new video (Figure 7 video-1) to show the changes of MaLionR sensor in HEK293 cells undergoing hypoxia and reoxygenation. Establishing hypoxia assay for imaging live HEK293 was more challenging than we expected, but we hope reviewers find the data in this manuscript satisfying.

2. We revised and streamlined the Discussion, making it ~200 words shorter and more concise.

3. We also reformatted the manuscript and added additional files such as raw data to meet *eLife* requirements.

Reviewer #1 (Evidence, reproducibility and clarity (Required)):Summary:This manuscript takes a closer look at how hypoxia affects the accumulation of PI4P and PI4,5P2 (PIP2) in the plasma membrane of *Drosophila* ovarian follicular epithelial cells and how ATP depletion similarly affects the localization of the same phospholipids in HEK293 cells. They demonstrate that hypoxia results in the reversible loss of plasma membrane (PM) association of both lipids, with PIP2 disappearing ahead of PI4P, and recovering more slowly than PI4P when oocytes are returned to normoxia. They also show that the intracellular vesicular pools of PI4P are depleted ahead of the PM pools and the PI4P recovery occurs first in PM, then in the vesicles. They show that the disappearance and recovery of the polarity protein Lethal giant larvae (Lgl) parallels that of PIP2 during hypoxia and subsequent normoxia, with a very slight delay. The authors then go on to show the RNAi knockdown of the PM enzyme (PI4KIIIa) that phosphorylates PIP delays the recovery of PI4P at the membrane, with recovery first occurring in the vesicular pools. This knockdown also delays the recovery of PIP2 and, as with recovery of PI4P, the recovery of PIP2 now occurs first in vesicular pools. Lgl recovery follows that of PI4P and PIP2 with RNAi knockdown of PI4KIIIa. The knockdown of all three of the enzymes that phosphorylate PIP to generate PI4P delays recovery of PI4P, PIP2 and Lgl at the membrane even more. The authors show that proteins required for the PM localization of PI4KIIIa have similar effects on the recovery of PM PI4P, PIP2 and Lgl (with delays and recovery of vesicular pools before PM pools). Independently, the authors show that ATP depletion in HEK293 cells result in similar reversible depletion of PI4P, PIP2 and Lgl from the PM. From these studies and their previous findings, the authors conclude that pools of PI4P and PIP2 are likely rapidly turned over in the membrane even during normoxia and that this rapid recovery is dependent on the PM localized enzyme that phosphorylates phosphoinositol.Major comments:Overall, the data are beautifully presented; it is quite helpful to have a video of each experimental treatment showing the corresponding response of all three molecules that are being monitored. Signal quantification over time is carefully documented. With the exception that a link between hypoxia and depletion of ATP has not been demonstrated here, the key conclusions are convincing. However, as pointed out below (in the significance section), some of the major points have already been published by this group. Their conclusion that hypoxia induces acute and reversible reduction of cellular ATP levels (which are then proposed to affect the activities of the enzymes required for PI4P and, consequently, PIP2 production) was not shown. They did demonstrate that acute depletion of ATP had the same consequences on PM phospholipids as acute hypoxia (in HEK293 cells). And, indeed, it makes sense that hypoxia could affect enzymes required for ATP synthesis, but the authors would have to show that acute hypoxia results in acute reduction in cellular ATP pools to make the links they suggest. This is something they should be able to do in the HEK293 cells now that they have their ATP sensor. Just to note, this group did show that hypoxia can reduce levels of ATP in *Drosophila* oocytes in their previous paper (Dong et al., 2015, Figure S3), but it is unclear if this is reversible and happens in the time frame of the experiments presented in this current manuscript.

We appreciate reviewer’s point on our previous studies on hypoxia and ATP inhibition. In *Dong et al. 2015* we biochemically measured ATP levels in embryos treated by hypoxia, but due to lack of ATP biosensors it was not possible then to show real time ATP level changes in cells undergoing hypoxia and reoxygenation. Instead, we showed that direct ATP inhibition by antimycin treatment mimics the effect of hypoxia, supporting the hypothesis that hypoxia acts through ATP inhibition.

In the current manuscript, we are able to demonstrate for the first time that hypoxia triggered acute and reversible ATP level reduction in *Drosophila* follicular epithelial cells (Figure 7—figure supplement 1A). Furthermore, we are able to show the close correlation of ATP and PM PI4P/PIP2 levels, and identified PI4KIIIa as one of the key enzymes in the process. In the finalized manuscript we also added data to show ATP level changes in HEK293 cells under hypoxia, as suggested by the reviewer (Figure 7—figure supplement 1B).

My suggestions are the following:(1) The authors need to make it absolutely clear what was already known, including the following: (A) hypoxia reversibly affects PM pools of PI4P, PIP2, and Lgl (and other membrane associated proteins), (B) that hypoxia can affect ATP levels in *Drosophila* oocytes (although these previous studies do not show anything about the dynamics) and (C) that reducing ATP levels affects PM pools of PI4P, PIP2 and Lgl.

We agree with reviewer and have revised the introduction (p3, second paragraph) to better summarize what we previously observed on hypoxia/ATP and PIP2 turnover. It should be noted though that our previous studies did not contain any data regarding PI4P changes under hypoxia or ATP inhibition, as the current manuscript is the first time we reported the making and use of PI4P sensor such as P4Mx2::GFP in *Drosophila*.

(2) They should demonstrate that acute hypoxia and return to normoxia has acute and reversible effects on cellular ATP levels – they now have the tools to do this, at least in HEK293 cells.

We agree with reviewer and are happy that we are able to add this data to the final revision. Such experiments did require significantly modified setup for imaging live HEK293 cells with controlled hypoxia/reoxygenation and we had to spend more than a month to optimize the experiments. The data are presented in Figure 7—figure supplement 1C

Minor comments:The manuscript is too long and the discussion unnecessarily repeats everything already presented in the results. The authors should find a way to streamline the discussion.

We revised the final manuscript and make the discussion ~200 words shorter and more concise. Paragraphs rephrasing the results were removed.

N values should be given for all figures and experiments, and the N=23/24 versus N=24/24 needs to be explained the first time it is used.

We have revised manuscript so all N values are clearly provided and easier to understand.

There are a few mismatches in terms of plural nouns and singular verbs and vice versa sprinkled into the manuscript, so some careful editing would be useful.

We have revised the manuscript to eliminate such errors/typos, especially with the help of the generous and comprehensive list of by reviewer#3.

Significance:I was initially quite excited about the novelty of their findings and the potential insight into the dynamics of PM pools of the two phospholipids that are critical to cell polarity and that play important signaling roles. However, at least a subset of their conclusions were either published in their earlier work or do not necessarily follow from what they have done in this manuscript. Their statement that hypoxia in *Drosophila* induces acute and reversible depletion of PM PI4P and PIP2 was presented in a previous publication (See Figure 8 of Dong et al., 2015).

We greatly appreciate reviewer’s comments on the significance of our discovery. Again all data regarding PI4P are new in this manuscript and have not been published before. We only published very preliminary data suggesting the reversible depletion of PIP2 and PIP3 (but not PI4P) under hypoxia (Dong et al., 2015). The current manuscript provides a comprehensive set of quantitative live imaging data with high spatial and temporal resolution that demonstrate for the first time the dynamic turnover of PM PI4P under hypoxia and ATP inhibition, the correlation between such turnovers of PM PI4P and PIP2, and the direct correlations between PI4P/PIP2 turnover and Lgl electrostatic PM targeting and intracellular ATP levels. In addition, studies on the role of PI4KIIIa complex in such process have not been done before.

This manuscript would appeal to an audience interested in the mechanisms of cell polarity and phosphoinositide signaling.I am a *Drosophila* developmental geneticist quite familiar with the topics that this paper addresses.Reviewer #2 (Evidence, reproducibility and clarity (Required)):Summary:This manuscript describes the effect of hypoxia on the levels of PI4P and PI45P2 , two key PPIs that are enriched on the inner leaflet of the plasma membrane. These PPIs are synthesized by the sequential phosphorylation of π by a PI-4 kinase and subsequently a PI4P 5 kinases, both of which use ATP. The relevant PI-4 kinase at the plasma membrane, PI4KIIIa has been conclusively identified previously in mammalian cells by the DeCamilli lab (Nakatsu et.al JCB 2012) and its role in regulating the synthesis of PI4P and PI(4,5)P2 in two Drosophila cell types in vivo shown by two previous studies. Balakrishanan et.al J.Cell Sci 2018 (photoreceptors during PLC signalling) and Basu et.al Dev.Biol 2020 ( in multiple larval cell types ). PI4KIIIa has been shown to exist as a complex of the enzymatic polypeptide, EFR3 and TTC7. The studies by Nakatsu, Balakrishnan and Basu have shown the importance of the complex subunits is regulating PI4P and PI45P2 levels in cultured mammalian cells and *Drosophila* cell types in vivo.

We thank reviewer for pointing out the work of Balakrishanan et.al 2018. We have added a brief summary this reference to the Discussion in revised manuscript (p13, line 10-12)

In the present study, Lu et. al build on their previous work showing that the polarity protein Lgl undergoes hypoxia induced translocation. They show that hypoxia also induces loss of PI4P and PI45P2 at the plasma membrane in these cells correlated with loss of Lgl localization to the PM. The manuscript then goes on to establish the requirement of the PI4KIIIa complex in regulating Lgl localization as well as PI4P and PI45P2 levels at the plasma membrane during hypoxia and the subsequent recovery of these at the plasma membrane.The strength of the manuscript is twofold.(i) The work is done to a high technical standard and the investigators have carried out the measurements of LGL localization, PI4P and PI45P2 levels along with simultaneous measurements of ATP levels in vivo. The work would be strengthened further if the authors could show the level of depletion of PI4K isoforms or PI4KIIIa complex subunits units induced in ovarian tissue under their experimental conditions by the GAL4 drivers used in this study. This is not a persnickety detail as RNAi lines can have very different effectiveness in *Drosophila* ovarian tissue compared to other fly cell types. This point is, in particular, important in cases where an RNAi line is being used and the conclusion is a lack of impact on a phenotype being studied.

We are fully aware of the potential caveat of RNAi. In our previous publications we were able to validate RNAi knock-down efficiency against endogenously or ectopically expressed GFP-tagged target proteins (Dong et al., 2020; Dong et al., 2015; Lu et al., 2021) or endogenous proteins with available antibodies (Dong et al., 2015). It is regrettable that presently such reagents are not available for directly examining the level of RNAi knock-down for PI4KIIIa and PI4KIIa etc. We did show that *rbo-RNAi* efficiently knocked down the expression levels of Rbo::GFP (Figure5—figure supplement 1C). In current manuscript, we have been very careful to draw conclusions based on negative RNAi results.

(ii) A second strength is that the authors now illuminate a further in vivo cell type where the function of the PI4KIIIa complex in regulating PI4P and PI45P2 levels. This adds to the earlier work of Nakatsu, Balakrishnan and Basu.A key difficulty with the current story is the lack of specificity of the phenotype they demonstrate under hypoxia. Of course, hypoxia is expected to deplete cellular ATP levels but PI4KIIIa is not the only enzyme that this lack of ATP will impact. There will be dozens or more other kinases, both protein and lipid kinases whose function will be impacted by the drop in ATP levels. Therefore, it is hard to attribute a specific/particular role to the PI4KIIIa complex under these conditions. The mislocalization of LGL::mCherry while correlated with PI4P and PI45P2 levels at the plasma membrane may be just that- a correlation. It is quite possible, indeed likely, that the mislocalization of LGL-mCherry under hypoxia conditions is due to the reduction of the activity of another lipid or protein kinase due to the drop in ATP levels due to hypoxia (PKC is a possibility too).

We agree with reviewer that PI4KIIIa almost certainly is only one of the enzymes that are involved in regulating the PM PI4P/PIP2 turnover triggered by hypoxia. This manuscript is our first effort to investigate the potential regulatory network underlying the hypoxia-triggered turnover of PM PI4P and PIP2, and it is our long term goal to identify more components in the regulatory network.

As to underlying mechanisms of the loss of PM Lgl under hypoxia, we previously showed that PM targeting of Lgl dependents on both PI4P and PIP2 and acute depletion of PI4P and PIP2 in cultured cells completely blocks the PM targeting of Lgl (Dong et al., 2015). Thus, although we cannot exclude contributions from other lipids, it is highly plausible that loss of PM PI4P and PIP2 triggered by hypoxia is the main driving force disrupting the electrostatic PM targeting of Lgl.

Lgl is phosphorylated by aPKC and such phosphorylation inhibits Lgl PM targeting by neutralizing the positive charges on Lgl’s polybasic motif (Dong et al., 2015, Bailey et al., 2015). Thus, potential inhibition of aPKC activity by hypoxia should not inhibit the PM targeting of Lgl. Consistently, we previously showed that *aPKC*^-/-^ mutant cells showed same acute and reversible loss of PM Lgl under hypoxia (Dong et al., 2015).

Minor comments:The authors must reference all published work on the PI4KIIIa complex in the literature. Some of it is excluded in the present version

We apologize for the missing references and in the revised manuscript we have already added several additional references based on the suggestions of reviewer#1 and #3. In the finalized manuscript we had made our best effort to cover all the relevant studies.

The *Drosophila* work, particularly cell types used, etc are not accessible to people who are not fly experts. This should be done.

We added a sentence to the end of the first paragraph in Results to specifically highlight that all *Drosophila* studies were based on follicular epithelial cells from female ovary (p4, line 25-27).

Significance:Adds to knowledge on the PI4KIIIa complex.Builds on existing knowledge in the PI4KIIIa field and maybe also cell polarity field.Reviewer #3 (Evidence, reproducibility and clarity (Required)):Summary:Phosphatidylinositol phosphates (PIPs) are key determinants of membrane identity and regulate crucial cellular processes such as polarization, lipid transfer and membrane trafficking. Despite decades of study, surprisingly little is known about how levels of PIPs are regulated in response to cellular stress. Here, using *Drosophila* ovarian follicular epithelial cells and human HEK293 cells, the authors show that levels of plasma membrane (PM) PI4P and PIP2 decrease rapidly in response to hypoxia, resulting in loss of polybasic proteins from the PM. These effects are reversed in response to reoxygenation. Similarly, hypoxia leads to acute depletion of ATP levels, which also regenerate following reoxygenation. Using a combination of quantitative image analysis and genetic analysis, they show that PI4KIIIalpha and its binding partners Rbo/ EFR3 and TTC7 are needed to maintain PI4P and PIP2 at the PM under normal and hypoxic conditions, whereas the other two Drosophila π 4-kinases, Fwd/PI4KIIIbeta and PI4KII, play a less important role in PM PIP homeostasis. Their results suggest that manipulations with indirect effects on PIPs (hypoxia, ATP depletion, ischemia) can have a profound impact on electrostatic charge at the PM, as well as downstream processes that require PM PI4P and PIP2.Major Comments:1. In general, the authors' conclusions are convincing. However, some of the results are less evident from the still images and graphs provided in the figures than from the videos that accompany the figures. Some suggested improvements are below.2. No additional experiments are essential to support the claims of the paper, although some additional quantitation would be helpful to the reader, as detailed below.3. Data and methods are generally presented in such a way that they can be reproduced, although some additional details would be helpful, as listed below.4. Experiments were adequately replicated, and statistical analysis appears adequate.

We are extremely grateful to the generous efforts of the reviewer providing such a detailed list of suggested improvements. We have incorporated all the text revisions into the revised manuscript and revised the figures accordingly too.

Minor comments:1. Although the data are generally quantified quite well, there are two instances in the first full paragraph on p. 5 where this is not the case. First, PM PI4P is described as "oftentimes" as showing a transient increase in the early phase of hypoxia. However, this is not quantified. How often did this occur among the samples examined? How large is the transient increase when it occurs (Figure 1A' error bars are not obvious on the colored background)? Second, the authors state that the P4Mx2-GFP puncta "often" became brighter after recovery. How often did this occur? No quantitation is provided.

Upon close inspection of the data, we conclude that during the early phase of hypoxia PM P4Mx2::GFP always showed an initial drop followed a transient increase. Thus we revised the sentence to delete “oftentimes”.

We did not specifically quantify the transient increase of the PM P4Mx2::GFP during the early phase of hypoxia since it is likely an artifact as discussed in the manuscript, making its quantification less meaningful.

As to the P4Mx2::GFP puncta, regretfully we do not have imaging tools that can accurately and automatically recognize/measure such puncta in our live recordings. We are actively developing such software using trainable Weka segmentation tool (https://imagej.net/plugins/tws/) and hopefully such puncta quantifications will be possible in our future experiments.

2. The authors conclude that "PI4KIIalpha and Fwd contribute significantly to the maintenance of PM PI4P" (bottom of p. 7), yet they did not validate their RNAi knockdowns of these two genes, so they do not know whether it is one or both of these PI4Ks that contribute.

We agree with reviewer that our RNAi knockdowns on PI4KIIa and Fwd are not sufficient to tell whether one or both contribute to the PM PI4P maintenance. We revised the sentence to “Our data support that PI4KIIα and/or FWD contribute significantly to the maintenance of PM PI4P…” (p7, line 33-34)

3. In Figure 4B, a subset of the cells "show failed recovery of PM Lgl::GFP". However, some cells did recover. This average percentage of cells that recovered should be quantified, if possible.

Added numbers of PI4K-3KD cells that show normal or failed hypoxia response of Lgl::GFP and revised the sentences accordingly (p8, line20-23)

4. In Figure 7A, B, the bottom cell in each example lags behind the top cell in recovery of the MaLionR sensor. The frequency of observed cells in each class for 7A, B should be quantified.

Added n numbers of each cell class to Figure 7A, B legend.

5. In most cases, prior studies were referenced appropriately. However, two previous studies in Drosophila showing the effects of Sktl/PIP2 reduction on localization of polybasic proteins Lgl, Baz/Par-3 and Par-1 were not cited (relevant to the first paragraph of the Introduction, p. 3): Gervais et al., Development (2008), Claret et al. Curr Biol (2014). In addition, two studies showing the importance of *Drosophila* PI4KIIIalpha in synthesizing PM PI4P and PIP2 were not cited (relevant to the description of this enzyme, top of p. 6): Yan et al., Development (2011), Tan et al., J Cell Sci (2014). Data showing fwd null mutants are not lethal (relevant to top of p. 7) were published in Brill et al., Development (2000).

We thank reviewer for suggesting additional references. We added *Yan et al.* and *Tan et al.* for referencing PI4PIIIa, and *Brill et al.* for referencing the original characterization of fwd. We discussed work from *Gervais et al.* and *Claret et al.* in the revised discussion (p6, line 11-16).

6. For the most part, text and figures are clear and accurate. However, there are quite a few typos and grammatical mistakes, as well as instances of lack of clarity in the writing that should be addressed. In addition, there are a number of improvements to presentation of data that would make the figures easier to understand. These are listed below.7. Suggestions to improve presentation of data and conclusions are below.

Again, we greatly appreciate such generous efforts from the reviewer and have incorporated all the text revisions into the revised manuscript.

Significance:Overall, the authors do a nice job of showing that hypoxia leads to previously unappreciated effects on levels of PM PI4P and PIP2, resulting in loss of PM association of proteins important for normal cellular physiology. This finding is quite novel. Moreover, the authors provide insight into the identity of the PI4Ks that are responsible for regenerating PM PIP2 following return to normoxia. Their analysis of the dynamics of these changes provides multiple interesting insights, including the potential roles of intracellular pools of PI4P in replenishing PM PIP2 and the observation that intracellular accumulation of PIP2 is occasionally observed in association with the appearance of intracellular PI4P puncta, suggesting a novel route for PIP2 replenishment in response to hypoxic stress. Their results will provide the basis for future studies examining the cellular mechanisms involved. This study will be of interest to those studying phosphoinositide biology as well as cellular responses to hypoxic stress and recovery, such as occur during ischemia and reperfusion.Reviewer expertise: *Drosophila* molecular genetics, cell biology, developmental biology, phosphoinositides, PIP pathway enzymes, PIP effectorsReferees cross-commentingThis session includes the comments of all reviewers.Reviewer 3: I agree with reviewer #1 that the authors did not do a good job of clarifying what they and others had previously shown, and I must confess I didn't carefully examine their previous papers carefully enough before preparing my review. In fact, they previously showed that hypoxia affects localization of Dlg at the plasma membrane and that its recovery depends on PI4KIIIalpha and PIP2 (Lu et al., Development 2021). This is in addition to their previous data showing effects of hypoxia on Lgl (Dong et al., J Cell Biol 2015). Thus, less of the information in the current manuscript is novel than I thought when I initially read it.I also agree with reviewer #2 that they need to do a better job of citing the relevant literature and considering the possibility that hypoxia and reduced levels of ATP might affect many different enzymes. In addition, as suggested by reviewer #1, it seems importantReviewer 1: I agree with what Reviewer 3 is suggesting and with reviewer 2 that the authors should do a better job of citing all of the relevant literature. I also appreciate the detailed edits provided by Reviewer 3 – it was very generous of them to do this.Reviewer 2: The points raised by reviewer 1 and 3 with regard to the citing or prior work (from the authors or other labs) also applies to their citing of literature on π and PI4K signalling. Here too citing or prior work has been less than satisfactory making it difficult to do this.

We want to thank all three reviewers for their thoughtful and constructive comments. We have revised the introduction to better summarize what we had observed in our previous studies. On the other hand, this manuscript presents a systematic study on the hypoxia-triggered turnover of PM PI4P and PIP2, the correlation between PI4P/PIP2 turnover and electrostatic PM targeting of Lgl, as well as a potential role of PI4KIIIa and its PM targeting mechanism in regulating the turnover of the PM PI4P and PIP2 under hypoxia. Although the latter by no means indicates that PI4KIIIa is the only enzyme in regulating such process, its characterization is the beginning for us to further identify additional enzymes and regulators in this hypoxia triggered phenomenon.

We have added additional references as suggested by the reviewers in the revised manuscript, and made our best efforts to have all relevant references cited.

Description of the revisions that have already been incorporated in the transferred manuscript

(Note: *Current resubmission is formatted to meet eLife style. To avoid confusions, we kept the figure references etc as the same in the original manuscript*)

We have incorporated nearly all of the suggestions from reviewer#3 into the current revision, with few exceptions as listed at the end of this letter. Below are point-to-point responses to selected suggestions involving data interpretation and comprehensive text revisions

p. 5, first paragraph, line 2: replace "oftentimes" with "often" and provide quantitation (see above)

Deleted the “oftentimes”. Upon close inspection of our data we conclude that PI4P always showed transient increase of PM signal in early hypoxia.

p. 5, first paragraph, line 6: the claim of "often" should be quantified (see above)

Deleted the “often”. PI4P puncta actually were consistently brighter after recovering from hypoxia.

p. 5, second paragraph: the extent of recovery of Lgl is less when Lgl-RFP is coexpressed with PLC-PH-GFP, potentially due to titration of PIP2 by PLC-PH; the authors should comment on this

This is a good suggestion from the reviewer. Revised by adding to the end of paragraph: *“Note that in Figure 1B Lgl::RFP recovery appears lower than in wild type, possibly due to the titration of PIP2 by PLC-PH::GFP expression.”*

p. 5, last line: the authors should provide information about the "targeted RNAi screen"; which genes were tested? did any others give relevant phenotypes? a table showing the results of the screen should be provided as supplementary information

Added Table S1 which summarizes the results of RNAi screen.

p. 11, first full paragraph, line 6: what about PI4KIIIbeta? is the KmATP for this enzyme known?

Based on literature, KmATP of PI4KIIIbeat is similar to PI4KIIIa’s (~400uM, Balla and Balla 2006). We added the PI4KIIIb KmATP value to the revised discussion (p11, line25)

p. 11, last paragraph, line 2: what is meant by "etc." is unclear; remove "etc." and include specific information related to what was reported in the literature (with proper references)

Revised the sentence to “*…that KmATP of PI4KIIIα was measured decades ago using purified PI4KIIIα enzymes from tissues such as bovine brains and uterus (Carpenter and Cantley, 1990)*”. The reference (Carpenter and Cantley, 1990) is a review which contains details of biochemical characterizations of PI4K kinases from numerous publications.

p. 12, line 3: why do the authors claim that the intracellular pool of PI4P is first synthesized by PI4KIIalpha? what about PI4KIIIbeta? their results do not distinguish between these enzymes

We favor the hypothesis that PI4KIIa is responsible for synthesizing the intracellular pool of PI4P because the very low KmATP of PI4KIIa. PI4KIIIbeta has high KmATP similar to PI4KIIIa.

p. 12, last paragraph, lines 6-7: for the reader, please clarity the mechanism that was invoked to explain how PIP5K can make PIP2 from π in *E. coli* (Botero et al., 2019)

Revised the sentence to *“PIP5K can be sufficient to make PIP2 from π in E. coli by phosphorylating its fourth and fifth positions (Botero et al., 2019) in the absence of PI4Ks*”

p. 13, first paragraph, last line: cannot conclude that components of PI4KIIIalpha are "highly interdependent" without testing effect of knockdown of PI4KIIIalpha on Rbo and TTC7, etc.; instead, can conclude that the data are consistent with all of the components acting in the same process; also, delete "the" before "proper"

Revised the sentence to “*.. supporting that components in PI4KIIIαa complex may act interdependently for the proper PM targeting* in vivo*.*”

p. 14, second paragraph, lines 3-5: expand on this idea; what additional lipids could be important here? are there examples of other proteins that require these additional lipids?

We revise the sentence to “*The reason for such difference between Lgl::GFP and PLC-PH::GFP in rbo-RNAi cells is unclear, though likely derives from the requirement of polybasic motif proteins for additional anionic lipids at the plasma membrane, specifically high mol fractions of phosphatidylserine in addition to lower concentrations of polyanionic phosphoinositides (Yeung et al., 2008).* “

p. 16, line 6: explain in brief what "pNP plasmid" is and how the multi-RNAi method works (what promoters drive expression of the shRNAs, how many shRNAs are included in the plasmid, etc.)

Added a section in Material and Methods to describe the details of the generation of pNP constructs and fly stocks.

p. 16, lines 8-3: appropriate references should be included for each stock where available

Added references to stocks cy2-Gal4, rbo::GFP and UAS-AT1.03NL1.

p. 16, line 11: explain what UAS-AT1.03NL1 is

Added: “*UAS-AT1.03NL1(DGRC#117011) was used to express the Drosophila-optimized ATeam ATP sensor AT[NL] in follicle cells (Tsuyama et al., 2017)*”

p. 16, lines 16-17: Gerry Hammond should not be listed as providing these constructs if he is a coauthor on the manuscript; appropriate references should be cited for these constructs

Revised the sentence to “*Mammalian constructs of P4M::GFP, P4Mx2::GFP, and PLC-PH::GFP were as previously described (Hammond et al., 2014; Hammond et al., 2012; Várnai and Balla, 1998).*”

p. 17, lines 3-4: sentence fragment "Images were further" is not complete

This was a typo, deleted.

p. 19, lines 5-6 from bottom: title doesn't accurately reflect that PI4P doesn't appear to recover in WT control; why is this the case? recovery was observed in other experiments

Live recording showed that PM PI4P did recover during reoxygenation (Figure 3A, Video S7). This particular recording in Figure 3A/Video S7 was a bit difficult for automatic quantification by our custom software due to that P4Mx2::GFP signal somehow was weak and noisy, resulting in less than “ideal” recovery curves.

p. 22, line 16: fix typo in "uncalibrated"; spell out what "AT[NL] sensor shows

Revised to “*Heat map of the (uncalibrated) FRET ratio of ATeam ATP sensor AT[Nl] in follicle cells of a dissected ovary undergoing hypoxia and reoxygenation* ex vivo”

Reviewer#3’s suggestions to improve the figures and videos:- replace colored labels on black boxes with colored labels on white background (Figure 5A (left), Figure 5B-D (top), Figure 6A (left), Figure 7A-C (left), Figure S1A (left), Figure S1C (top), Figure S2A (left), Figure S4 (left))

We have revised the figures accordingly.

- provide scale bars throughout (Figures2-7, S1-S4)

We have revised the figures accordingly.

- provide vertical lines similar to those in Figure 4A' in all of the time-course graphs and/or making the background colors slightly darker (Figures2A', 3A', 5A', 6A'); also make the error bars darker (Figure 1A'-C', Figure 4, Figure 5A', Figure S2B)

We revised all backgrounds in charts to make them similar to Figure 4A’.

- for consistency, label PM index graphs in Figure 4 and Figure S2 as Figure 4A' and Figure S2A'

We revised figures 4 and S2 accordingly.

- why are some of the PM index graphs labeled "PM index" and others labeled "PM index-1" on the Y-axis? this should be explained or changed for consistency

The mixed use of “PM index” and “PM index-1” were relics due to different versions of software used throughout the project. We revised all graphs to make Y-axis “PM Index” label consistent.

- "blank diamonds" described in figure legend for Figure 7B' are barely visible when printed

Revised Figure 7B’ by filling blank diamonds with grey color to increase their visibility.

- Figure 7C is mislabeled (MaLionR label should be replaced with PLC-PH-RFP)

We corrected this error in revised Figure 7C.

- in Figure S3A, it would help to know the size of the cells (i.e., how many were present in the area examined)

Revised the Figure S3A legend to clarify that each circle covers approximately three to four cells.

- in Video S19, "PLC-PH::RFP" is mislabeled "PLC-PH::GFP" (both P4MX2 and PLC-PH are labeled GFP in the video)

We renamed the video file to correct this typo.

Description of analyses that authors prefer not to carry out

(Note: *Current resubmission is formatted to meet eLife style. To avoid confusions we kept the figure references etc as the same in the original manuscript*)

Reviewer#2: The work would be strengthened further if the authors could show the level of depletion of PI4K isoforms or PI4KIIIa complex subunits units induced in ovarian tissue under their experimental conditions by the GAL4 drivers used in this study.

We are regretful that we are unable to directly evaluate the RNAi knock-down efficiency of several genes such as PI4KIIIa. We have nonetheless been careful to draw the conclusions in the manuscript in accordance with the potential caveat of RNAi experiments. We did directly show that rbo-RNAi directly knocked down the Rbo::GFP (Figure S1C). In addition, although we could not confirm the knock-down of ttc7-RNAi, we showed it can reduced level of Rbo::GFP, which is likely due to an effective knock-down of TTC7 (Figure 6B).

Reviewer #3:p. 6, second full paragraph, lines 6 and 9: the callouts should be to Videos S5, not Video S4p. 7, second paragraph, line 3: change name of Video S7 to S6, and call out Video S6 herep. 7, third paragraph, line 2: change name of Video S8 to S7, and call out Video S7 herep. 8, first paragraph, line 8: change name of Video S6 to S8, and call out Video S8 here- Videos should be referred to in order (current Videos S7 and S8 should be renamed S6 and S7, and current Video S6 should be renamed S8)

We appreciate reviewer’s suggestion but decided to keep the videos in current order. Current resubmission is formatted to meet *eLife* style and each videos follow the main figures, which is easy for the readers to follow.

Reviewer #3: – replace pale colored boxes under labels for "hypoxia" and "air" with slightly darker boxes (Figure 1A-C, Figure 2A, Figure 3A, Figure 5A', Figure 6A', Figure S2B)

We tested many different combinations of colors and the current set appears to give the best contrast so far. We decided to keep the original color.

Reviewer #3- show single-color images in grayscale, which is easier to see on black and helpful for colorblind readers (applies to all figures except Figure S3); videos and merged still images should be shown in green and magenta for colorblind (not sure if channels in videos are difficult to change)

We converted Figure 1A to gray scale but found it visually inferior to the color version, as the gray images make the temporal differences between green and red channels much less pronounced. We have converted red channel in all videos to magenta color and also added text labels for each channel to reduce potential confusions since the manuscript contains total of nineteen video. Regrettably, to convert red channels in figures requires a very laborious process to recapture, recrop and recompose all the frames used in all figures. We hope that reviewer understand our decision to keep colors in figures unchanged.